# Identifying Insects, Clouds, and Precipitation using Vertically Pointing Polarimetric Radar Doppler Velocity Spectra

Christopher R. Williams[1], Karen L. Johnson[2], Scott E. Giangrande[2], Joseph C. Hardin[3], Ruşen Öktem[4,5], and David M. Romps[4,5]

[1] Ann and H.J. Smead Aerospace Engineering Sciences Department, University of Colorado Boulder, CO, 80309, United States
[2] Brookhaven National Laboratory, Upton, NY, 11973, United States
[3] Pacific Northwest National Laboratory, Richland, WA, 99354, United States
[4] Department of Earth and Planetary Science, University of California, Berkeley, CA, 94720, United States
[5] Climate and Ecosystem Sciences Division, Lawrence Berkeley National Laboratory, CA, 94720, United States

*Correspondence to*: Christopher R. Williams (Christopher.Williams@colorado.edu)

**Abstract.** This study presents a method to identify and distinguish insects, clouds, and precipitation in 35 GHz (Ka-band) vertically pointing polarimetric radar Doppler velocity power spectra and then produce masks indicating the occurrence of hydrometeors (i.e., clouds or precipitation) and insects at each range gate. The polarimetric radar used in this study transmits a linear polarized wave and receives signals in collinear (CoPol) and cross-linear (XPol) polarized channels. The measured CoPol and XPol Doppler velocity spectra are used to calculate linear depolarization ratio (LDR) spectra. The insect-hydrometeor discrimination method uses CoPol and XPol spectral information in two separate algorithms with their spectral results merged and then filtered into single value products at each range gate. The first algorithm discriminates between insects and clouds in the CoPol Doppler velocity power spectra based on the spectra texture, or spectra roughness, which varies due to the scattering characteristics of insects versus cloud particles. The second algorithm distinguishes insects from raindrops and ice particles by exploiting the larger Doppler velocity spectra LDR produced by asymmetric insects. Since XPol power return is always less than CoPol power return for the same target (i.e., insect or hydrometeor), fewer insects and hydrometeors are detected in the LDR algorithm than the CoPol algorithm, which drives the need for a CoPol based algorithm. After performing both CoPol and LDR detection algorithms, regions of insect and hydrometeor scattering from both algorithms are combined in the Doppler velocity spectra domain and then filtered to produce a binary hydrometeor mask indicating the occurrence of cloud, raindrops, or ice particles at each range gate. Forty-seven (47) summer-time days were processed with the insect-hydrometeor discrimination method using U.S. Department of Energy (DOE) Atmospheric Radiation Measurement (ARM) program Ka-band zenith pointing radar observations in northern Oklahoma (USA). For these 47 days, over 70% of the hydrometeor mask column bottoms were within +/-100 meters of simultaneous ceilometer cloud base heights. All datasets and images are available to the public on the DOE ARM repository.

## 1. Introduction

The vertical structure of non-precipitating clouds plays an important role in the Earth's radiation balance. These clouds absorb longwave radiation emitted from the surface and reflect shortwave solar radiation back into space (Cess et al., 1990). The proportion of these two processes determines whether these clouds act as a net radiation sink or source in the Earth's radiation budget (Ramanathan et al., 1989). Vertically pointing cloud radars have been used for decades to quantify the extent to which non-precipitating clouds can be used as inputs to Earth radiation budget studies to understand cloud dynamics and cloud life cycles (Moran et al., 1998; Ackerman and Stokes, 2003).

In addition to measuring cloud properties, cloud radars are sensitive enough to detect individual insects within the radar volume (for overviews, see Riley, 1989, Westbrook et al., 2014; Nansen and Elliot, 2016). The field of radar entomology exploits this sensitivity by pointing polarimetric radar beams a few degrees off vertical and rotating the beam 360 degrees in azimuth to estimate insect population and migration direction (Drake et al., 2020). The field of radar meteorology has used polarimetric scanning radar observations to track insect flying direction and altitude outside of clouds (Mueller and Larkin, 1985) and to estimate gust-front motions ahead of convective cells because insects and small pieces of vegetation act as radar reflectors trapped within the strong boundary layer outflow (Klingle el al., 1987). Insects are considered clutter and unwanted signals in vertically pointing cloud radar observations. Two approaches have been used to identify insects in cloud radar observations: polarimeteric signatures and Doppler velocity power spectra signatures. Compared to more spherical hydrometeors (i.e., cloud droplets, raindrops, and ice particles), insects have asymmetrical shapes that produce large cross-polarization power return signals that enable insects to be identified with polarimeteric radar estimates including differential reflectivity and linear depolarization ratio. (Lohmeier et al., 1997; Khandwalla et al., 2001 and 2002; Martner and Moran, 2001). Also, large insects will have different radar cross-sections at different radar operating wavelengths due to the resonance or Mie scattering effects enabling insects to be detected in dual-wavelength radar observations (Khandwall et al., 2001 and 2002; Kollias et al., 2002).

Insects produce unique signatures in the Doppler velocity power spectra. An individual insect scatters as a single point target with a returned power confined to a narrow Doppler velocity range and to a single range gate (Bauer-Pfundstein and Görsdorf, 2007). In contrast, clouds and precipitation are composed of hydrometeor distributions containing different size particles with different velocities that are spread over several range gates leading to broader measured Doppler velocity power spectra extending over several range gates (Luke et al., 2008). The difference between insect and hydrometeor Doppler velocity power spectra signatures has been used to distinguish insect and hydrometeor peaks in Doppler velocity spectra (Bauer-Pfundstein and Görsdorf, 2007; Luke et al., 2008). In these studies, multiple peaks are first found in the spectra and then intelligent algorithms (Bauer- Pfundstein and Görsdorf, 2007) or neural network algorithms (Luke et al., 2008) were developed to classify peaks as the result of either insect or hydrometeor scattering. The method presented herein reverses the processing steps by first identifying and then removing insect signatures in the Doppler velocity spectra before estimating spectrum peaks.

Identifying and removing radar scattering from insects and other sources of "atmospheric plankton" (Lhermitte, 1966) has been a known problem in developing operational cloud products (Kollias et al., 2016). The U. S. Department of Energy (DOE) Atmospheric Radiation Measurement (ARM) program merges observations from multiple sensors (including radars, lidars, and ceilometers) to produce an estimate of hydrometeors (i.e., cloud particles, raindrops, ice particles) in the vertical column, called the Active Remote Sensing of CLouds (ARSCL) product (Clothiaux et al., 2000). ARSCL is a high temporal (~4 s) and vertical (~30 m) resolution operational product that primarily uses ceilometer cloud base and radar moments to classify all returns into one of three scattering regimes: hydrometeor-only scattering, clutter-only scattering (due to insects or another non-atmospheric artifact), and scattering due to a mixture of hydrometeors and clutter within the radar pulse volume. An approximate estimate of maximum clutter height is provided to an automated heuristic algorithm developed over two decades of experience producing the

ARSCL product at multiple radar sites. The results of the classification are reviewed. On rare occasions, the maximum clutter height is revised and the classification procedure is repeated.

Figure 1 shows one hour of ARSCL processed reflectivity from the DOE ARM Southern Great Plains (SGP) central facility on 31-July-2018 (ARM user facility, 2014). Figure 1a (top panel) shows ARSCL reflectivity for radar volumes classified as either hydrometeor-only or hydrometeor-plus-clutter with Fig. 1b (middle panel) showing ARSCL hydrometeor-only reflectivities. The black symbols represent ceilometer-derived cloud base, which is also an input to the ARSCL operational algorithm. Figure 1c (bottom panel) shows the hydrometeor mask produced using the algorithms discussed herein. The apparent fall streaks in the ARSCL hydrometeor-only product below 1.5 km are misclassifications of insect clutter. The misclassification of insect clutter as hydrometeors and the inefficient ARSCL processing steps were some of the reasons why DOE ARM sponsored this work to identify insect clutter with the aim of improving future ARSCL products.

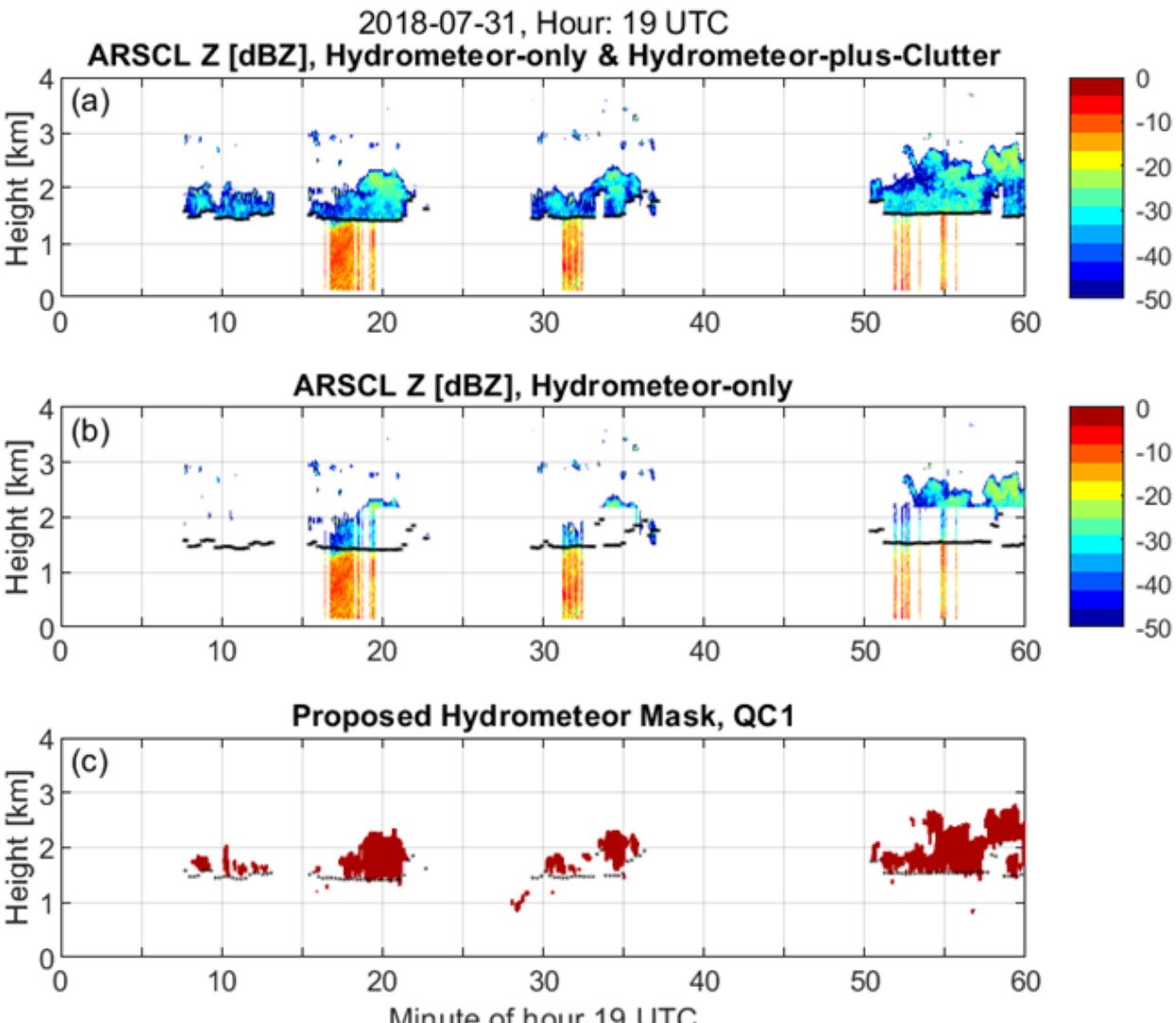

**Figure 1: Active Remote Sensing of CLouds product (ARSCL) for hour 19 UTC (hour 14 local) from the DOE ARM Southern Great Plains (SGP) central facility on 31-July-2018. (a) ARSCL reflectivity for radar volumes ARSCL classified as either hydrometeor-only or hydrometeor-plus-clutter. (b) ARSCL reflectivity for radar volumes ARSCL classified as hydrometeor-only. (c) Hydrometeor mask produced using the method described herein. Black symbols in all panels are ceilometer derived cloud base. Note the hydrometeor misclassification below ceilometer cloud base in (b) motivates the need for improved insect clutter detection.**

The method to identify insects and hydrometeors presented herein builds on prior work using polarimetric diversity and Doppler velocity power spectra variability (e.g., Martner and Moran, 2001; Bauer-Pfundstein and Görsdorf, 2007; Luke et al., 2008; Görsdorf et a., 2015). One unique feature of the proposed algorithms is that insect and hydrometeor scattering are identified before identifying significant peaks in the Doppler velocity spectra. This approach complements the methods that first identify multiple peaks and then classify each peak (Bauer-Pfundstein and Görsdorf, 2007; Luke et al., 2008) The observations used in this study and the signatures of insect and hydrometeor scattering are discussed in Sections 2 and 3. Section 4 presents the main concept behind the algorithms developed in this study. Section 5 compares the hydrometeor masks with the Clouds Optically Gridded by Stereo (COGS) product (Romps and Öktem, 2018) derived from stereo cameras. Section 5 also compares the hydrometeor mask cloud bottom with ceilometer derived cloud base. Conclusions and next steps are discussed in Section 6. The online Supplemental Material Section contains images of insect and hydrometeor classifications for forty-seven (47) summer-time days in northern Oklahoma, U.S.A., identified as LASSO cloud simulation events (LASSO, 2020).

## 2. Observations

The observations used in this study were collected by the U.S. Department of Energy (DOE) Atmospheric Radiation Measurement (ARM) program at their Southern Great Plains (SGP) central facility located in northern Oklahoma. Vertically pointing Ka-band radar co-polarized (CoPol) and cross-polarized (XPol) Doppler velocity power spectra are processed to identify insects, clouds, and precipitation in the vertical column. Verification of those classifications are based on observations from co-located lidar, ceilometer, Total Sky Imager (TSI), and cloud boundaries contained in the Clouds Optically Gridded by Stereo (COGS) product (Romps and Öktem, 2018).

*2.1 Ka-band ARM Zenith Pointing Radar (KAZR)*

The DOE ARM program deploys atmospheric observing systems to characterize the radiative properties of clouds in the atmosphere (Mather and Voyles, 2013). One of ARM's hallmark instruments is the Ka-band (35 GHz) ARM zenith pointing cloud radar (KAZR), which transmits linear polarized waves that are detected simultaneously with collinear polarized (CoPol) and cross-linear polarized (XPol) receivers. The received signals are processed to yield co-polarized $S_{signal}^{CoPol}(v_i, h_j)$ [Watts] and cross-polarized $S_{signal}^{XPol}(v_i, h_j)$ [Watts] Doppler velocity power at each velocity bin $v_i$ and range gate $h_j$. The linear depolarization ratio spectra profile $S_{dB}^{LDR}(v_i, h_j)$ [dB] is the ratio of polarized signal magnitudes defined as

$$S_{dB}^{LDR}(v_i, h_j) = 10log\left[\frac{S_{signal}^{Xpol}(v_i, h_j)}{S_{signal}^{CoPol}(v_i, h_j)}\right], \text{ or as} \tag{1a}$$

$$S_{dB}^{LDR}(v_i, h_j) = S_{signal,dB}^{XPol}(v_i, h_j) - S_{signal,dB}^{CoPol}(v_i, h_j) \tag{1b}$$

where $S_{signal,dB}^{XPol}(v_i, h_j)$ and $S_{signal,dB}^{CoPol}(v_i, h_j)$ are expressed in decibel units [dB] using $X_{dB} = 10log[X]$. The linear depolarization ratio $LDR$ [dB] is the integration of Xpol and CoPol signals over the spectrum and is defined as

$$LDR(h_j) = 10log\left(\frac{\sum_{v_{min}}^{v_{max}}\left[\frac{S_{signal}^{Xpol}(v_i, h_j)}{S_{signal}^{CoPol}(v_i, h_j)}\right]\Delta v}{\sum_{v_{min}}^{v_{max}}\Delta v}\right), \tag{2}$$

where $v_{min}$ to $v_{max}$ define the velocity range of valid $S_{signal}^{CoPol}(v_i, h_j)$ and $S_{signal}^{XPol}(v_i, h_j)$ observations.

At SGP, KAZR operates in the general 'GE' and medium 'MD' sensitivity modes to sense clouds at different altitudes with operating parameters during 2018 and 2019 shown in Table 1 (ARM user facility, 2011a, 2011b; Widener et al. 2012). Even

though insects are detected in both KAZR operating modes, to simplicity the figures and algorithm descriptions, the results from just the MD mode are presented herein. Since the MD mode transmits a long frequency-modulated pulse, the first resolved range gate is 570 m above the radar. The KAZR 3.05 m diameter Cassegrain parabolic reflector manufactured by Millitech produces a 0.2° antenna beamwidth with 57.5 dBi gain, has -27 dB cross polarization isolation, and a membrane radome slopes across the antenna with a dry 2-way loss less than 2 dB (Widener et al., 2012). The MD mode uses a non-linear frequency modulated chirp over a 3967 ns pulse length to produce a 45 m range resolution sampled at 30 m range spacing. At 1 km range, the radar pulse volume is a 3.6 m diameter horizontal disk over a 45 m range to yield a pulse volume of approximately 450 m$^3$. To save computer hard disk space, the KAZR CoPol and XPol Doppler velocity power spectra are retained only at range gates with significant power above a noise threshold.

*2.2 Validation Observations*

Two observational datasets are used to validate the derived KAZR insect and hydrometeor classifications: ceilometer cloud base estimates from a Vaisala model CL31 ceilometer (ARM user facility, 2010b; Morris, 2016) and cloud bottom and top estimates from the COGS product (ARM user facility, 2017). The Vaisala ceilometer uses a pulsed InGaAs diode laser at 910 nm wavelength and the vendor supplied algorithm estimates cloud base at 10-m and 16-s resolution when the vertical visibility is less than 100 m (Morris, 2016). The COGS cloud boundaries are derived from three pairs of stereo cameras positioned around the SGP Central Facility and represent cloud boundaries over a cubic domain 6 km to a side (Romps and Öktem, 2018). Due to camera visual occlusion during precipitation, COGS cloud boundaries are only estimated for cases of shallow cumulus clouds, which allow the three cameras to view the vertical extent of each cloud. Likewise, estimates from COGS are only available during daylight hours.

*Table 1.* **Operating parameters for KAZR deployed at ARM Southern Great Plains (SGP) during 2018 and 2019. Operating modes included General Purpose (GE) and Medium Sensitivity (MD) modes. Tabulated parameters include: pulse repetition frequency (*PRF*) [Hz], inter-pulse period (*IPP*) [μsec], number of points in FFT ($N_{FFT}$), number of averaged spectra (also known as number of incoherent integrations) ($N_{incoh}$), Nyquist velocity ($V_{Nyquist}$) [m s⁻¹], velocity resolution (Δv) [m s⁻¹], range to first range gate [m], range resolution [m], time-on target (which is calculated using $IPP\ N_{FFT}\ N_{incoh}$) [s], and time between samples [s].**

| Parameter | | |
|---|---|---|
| Sensitivity Mode | general 'GE' | medium 'MD' |
| Frequency [GHz] | 34.83 | 34.89 |
| Pulse Repetition Frequency (*PRF*) [Hz] | 2771 | 2771 |
| Inter-Pulse Period (*IPP*) [μsec] | 360 | 360 |
| Pulse Duration [ns] | 300 | 3967 |
| Pulse Modulation | none | Linear Frequency Modulation |
| Range resolution $\Delta R$ [m] | 45 | 45 |
| Distance between range gates [m] | 30 | 30 |
| Range to first range gate $R_1$ [m] | 100 | 570 |
| Number of points in FFT ($N_{FFT}$) | 256 | 256 |
| $V_{Nyquist}$ [m s⁻¹] | 5.96 | 5.95 |
| $\Delta v$ [cm s⁻¹] | 4.67 | 4.67 |
| Number of incoherent integrations $N_{incoh}$ | 20 | 20 |
| Time-on Target $t_{target} = IPP\ N_{FFT}\ N_{incoh}$ [s] | 1.8 | 1.8 |
| Time between samples $t_{sample}$ [s] | 3.7 | 3.7 |

## 3. Insect, Cloud Droplet, and Precipitation Spectral Characteristics

This section discusses the scattering characteristics of insects, atmospheric plankton, clouds, and precipitation as observed in KAZR CoPol and XPol Doppler velocity power spectra. The first subsection discusses characteristics when it is not raining and the radar is observing individual insects or other atmospheric plankton particles scattering as point targets with narrow velocity ranges and shallow cumulus clouds scattering as distributed targets with broader velocity ranges. The variability of return power across the Doppler velocity spectrum, or the spectrum 'texture', is used to distinguish point target insects from distributed target clouds. The second subsection describes the characteristics when individual insects and raindrop or ice particle distributions occur simultaneously in the radar volume. The LDR at each Doppler velocity bin is used to distinguish high LDR insects from low LDR raindrops or ice particles.

*3.1 Insects and Shallow Cumulus Clouds*

Figure 2 shows an hour of KAZR observations when insects (or other atmospheric plankton particles) and shallow cumulus clouds are observed over the radar during 1900 UTC (1400 Local Time) on 31-July-2018. From top to the bottom, Fig. 2 shows KAZR (a) CoPol reflectivity [dBZ], (b) mean Doppler velocity [m s⁻¹], (c) Doppler velocity spectrum width [m s⁻¹], (d) linear depolarization ratio (LDR) [dB], and (e) KAZR CoPol reflectivity at time-height locations (also called 'pixels' in this study) that do not have a LDR measurement. The black symbols in each panel indicate ceilometer-derived cloud base height, which is near

1.5 km for this hour. Below cloud base, reflectivity (Fig. 2a) and spectrum width (Fig. 2c) have a coherent pattern, but vertical motion (Fig. 2b) appears more variable. If drizzle or rain were below cloud base, then all three quantities would be coherent with downward motions increasing as reflectivity and spectrum width increase (Williams and Gage, 2009). Thus, it is not raining below cloud base. Above the ceilometer derived cloud base height, there are CoPol reflectivity observations (Fig. 2a), but not as many LDR estimates (Fig. 2d). For example, near minute 20, there is an enhancement of CoPol reflectivity above the ceilometer cloud base and extending above 2 km, yet, there are very few LDR observations in this time-height region. Since LDR requires both CoPol and XPol reflectivity observations, the lack of LDR above cloud base indicates that the XPol channel is not detecting cloud particles. This CoPol vs. XPol sensitivity is illustrated in the bottom panel which shows CoPol reflectivity for all pixels that do not also have a LDR observation. The continuous time-height CoPol reflectivity observations above 1.5 km are cloud features that are easily discernible by eye. Return signals from individual insects appear as speckles up to 4 km in all panels.

The CoPol and XPol Doppler velocity power spectra produced by individual insects and by cloud droplet distributions have different characteristics as illustrated in Fig. 3, which shows CoPol (Fig. 3a) and XPol (Fig. 3b) Doppler velocity power spectral density profiles at 19:19:02 UTC on 31-July-2018. The vertical axis extends from 0 to 3 km in height and the horizontal axis extends +/- 6 m s$^{-1}$ radial velocities. The Nyquist velocity is 5.95 m s$^{-1}$ and downward motions have positive values consistent with positive raindrop diameters having positive fall speeds due to gravity. Due to the long coded transmitted pulse, the first observations occur at 0.57 km range. The colors represent the return signal power expressed in dB with warmer colors indicating larger return signal power. The mean noise power is approximately -100 dB.

Figure 3c shows CoPol Doppler velocity power spectra at 1 and 2 km heights (black and red lines, respectively). The power spectrum at 1 km has more variability between velocity bins compared to the spectrum at 2 km. This variability is because the radar is detecting individual insects within the 300 m$^3$ field-of-view with each insect moving at its own radial velocity. If an insect is the only insect moving at a particular velocity, the spectrum will have an isolated peak (e.g., near -1.7 m s$^{-1}$ radial velocity in Fig. 3c). If multiple insects are moving at similar speeds, the spectrum will be broader, yet, will still have variability. For example, between -1 and +3 m s$^{-1}$ radial velocities, the 1 km height spectrum (black line) is both elevated in magnitude and has more bin-to-bin variability than the spectrum from 2 km (red line). Also, the backscattered power from insects is primarily confined to one range gate with some power leaking into neighboring range gates due to radar signal processing limitations, which produce point enhancements in the spectra profile. Shown in sequential spectra profiles in the Supplemental Material, point enhancements often appear in only one spectra profile and not in neighboring profiles separated 4 seconds apart. The surface wind speed was about 3 m s-1 for this profile and there is not enough information to determine whether the insects are passive tracers advecting with the wind or self-propelling themselves through the 3.6 m diameter by 45 m field-of-view in less than 4 seconds.

In contrast to individual insects, clouds and precipitation are distributed targets filling the radar volume with hundreds or thousands of hydrometeors of different sizes with different radial velocities. Since the number of particles in the hydrometeor size distribution varies gradually over neighboring particle sizes and the hydrometeor spectrum is extended in the velocity dimension due to antenna broadening effects, the return power spectrum has a gradual change over neighboring velocity bins. Thus, the power spectrum from a distribution of many hydrometeors is smoother than the return from a few individual insects. The smoother power spectra at 2 km height shown in Fig. 3c are consistent with a distribution of small cloud droplets moving at different velocities within the radar volume. In addition to smooth power spectra across the velocity dimension, power spectra from cloud droplets are also more continuous in range due to the vertical extent of clouds as can be seen with a continuity of clouds with height in Fig. 3a.

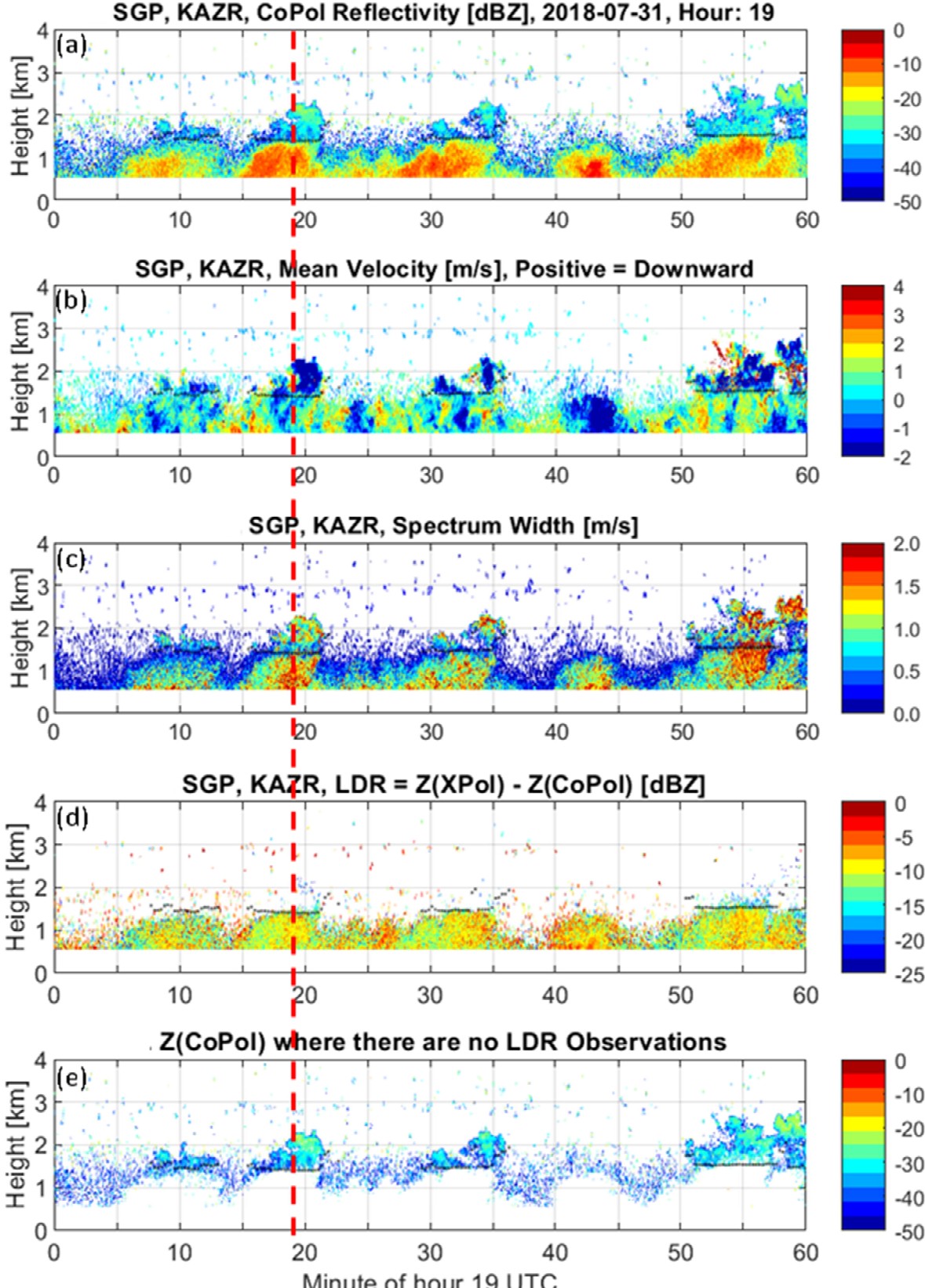

**Figure 2: Moments calculated from raw spectra for hour 19 UTC on 31-July-2018. (a) CoPol reflectivity [dBZ]. (b) Mean radial velocity [m s⁻¹], positive values are downward motion. (c) Spectrum width [m s⁻¹]. (d) Linear depolarization ration (LDR) [dB]. (e) CoPol reflectivity [dBZ] at pixels that do not have an LDR measurement. Black symbols in all panels are ceilometer derived cloud base. The vertical dashed line indicates time 19:02, which is the time of the profile shown figs. 3 and 6.**

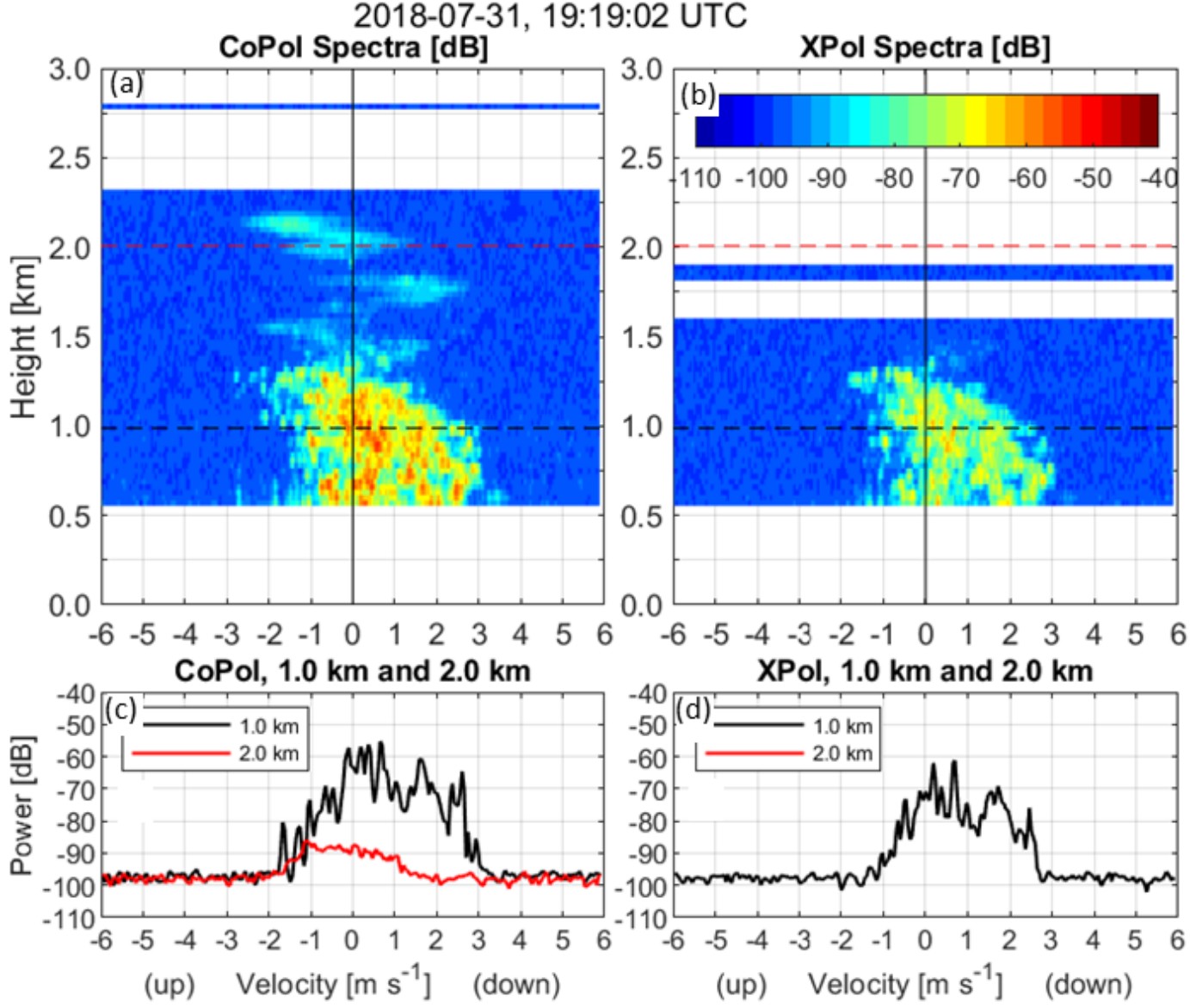

**Figure 3: Spectra from profile at 19:19:02 UTC on 31-July-2018. (a) CoPol Doppler velocity power spectra [dB] as a function of range and radial velocity. (b) XPol Doppler velocity power spectra [dB] as a function of range and radial velocity. (c) CoPol Doppler velocity power spectra at 1.0 km (black line) and 2.0 km (red line). (d) XPol Doppler velocity power spectra at 1.0 km (black line).**

5    *3.2 Insects and Precipitation*

Figure 4 shows time-height cross-sections of KAZR CoPol reflectivity (Fig. 4a) and LDR observations (Fig. 4b) when insects, clouds, and precipitation are observed in the same hour. Observations were collected during 0400 UTC (2300 Local) on 04-April-2019. From minutes 0-to-20, the approximate 1.5 km ceilometer cloud base height (black symbols) is above the insect layer that has LDR values between approximately -5 and -10 dB (see Fig. 4b), while the CoPol reflectivity is continuous in time and height

10    above the ceilometer cloud base height (see Fig. 4a and 4c). At the beginning of the hour, the CoPol reflectivity (Fig. 4a) time-height structure indicates a precipitating cloud system between 3 and 5 km that evolves in time with precipitation reaching the lowest resolved height of 0.57 km after minute 20. The LDR shows a similar time-height structure (with reduced vertical depth) with LDR values ranging between -25 to -20 dB. The LDR enhancement near 2.4 km and after minute 20 is due to a mixture of liquid and frozen particles near the melting layer (Baldini and Gorgucci, 2006). Below the melting layer, the LDR has values near

15    -25 dB that is due to scattering from rain drops. Above the melting layer, scattering from asymmetrical ice particles leads to LDR

values near -20 dB (Baldini and Gorgucci, 2006). In contrast to the shallow cumulus cloud droplet observations in Figs. 2 and 3, KAZR has enough sensitivity to detect XPol signal returns from large spherical raindrops and ice particles.

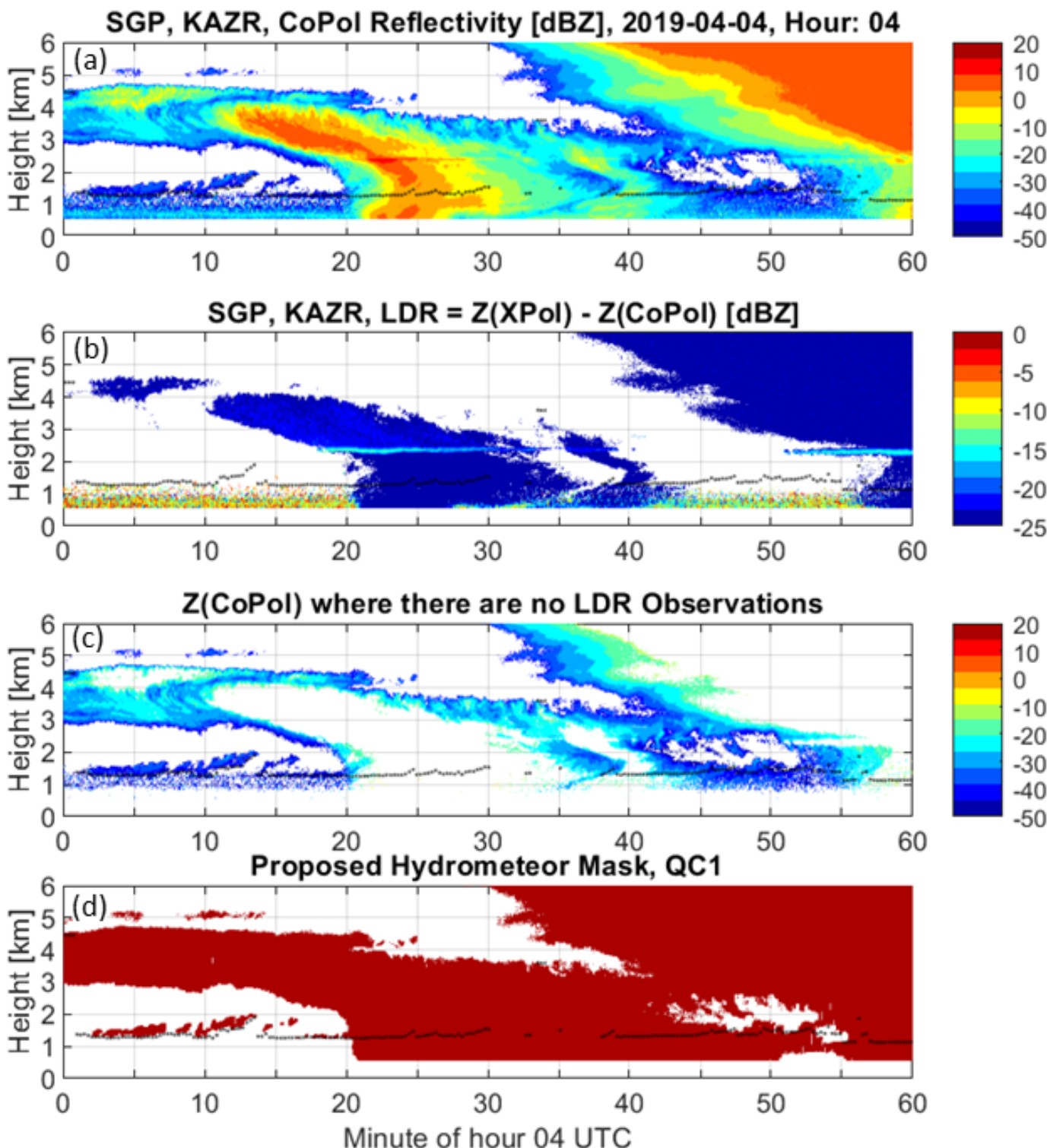

**Figure 4: Moments calculated from raw spectra and retrieved hydrometeor QC1 mask for hour 4 UTC on 4-April-2019. (a) CoPol reflectivity [dBZ]. (b) Linear depolarization ratio (LDR) [dB]. (c) CoPol reflectivity [dBZ] at pixels that do not have an LDR measurement. (d) Retrieved hydrometeor QC1 mask. Black symbols in both panels are ceilometer derived cloud base.**

Figure 4c shows the CoPol reflectivity at time-height pixels that do not have a LDR measurement. As with the shallow cloud observations (see Fig. 2e), there are more CoPol observations than LDR observations. The bottom panel (Fig. 4d) shows the QC1 hydrometeor mask produced by the insect-hydrometeor detection algorithm described in Section 4. The events shown in Figs. 2-4 highlight three important attributes of CoPol and LDR measurements:

- LDR measurements detect some, but not all, insect, cloud, and precipitation observations.
- KAZR LDR measurements do not have the sensitivity to detect shallow non-precipitating liquid clouds.
- Doppler velocity power spectra contain signatures unique to insect scattering and hydrometeor scattering.

The limitation of LDR measurements not detecting all insects detected by CoPol measurements and the benefit of Doppler velocity power spectra having signatures of insects and hydrometeor scattering suggests that Doppler velocity power spectra can be

analyzed along with LDR measurements to discriminate between insect and hydrometeor scattering.

## 4.    Anatomy of Insect-Hydrometeor Detection Algorithms

As described previously, the radar returned signal results from scattering from insects (including "atmospheric plankton") and/or hydrometeors (aka, cloud droplets or precipitation sized particles). The insect-hydrometeor detection algorithms described in this section aim to classify each region of the CoPol and LDR Doppler velocity spectra as either insect or hydrometeor scattering. Next,

the two CoPol and LDR regional spectral classifications are combined and then filtered to produce masks indicating the occurrence of insect or hydrometeor scattering at every range gate.

The detection algorithms start with the observed CoPol and XPol spectra profiles $S_{obs}^{CoPol/XPol}(v_i, h_j)$ [Watts]. These are a combination of signal power $S_{signal}^{CoPol/XPol}(v_i, h_j)$ [Watts] and random noise power $n(v_i, h_j)$ [Watts]

$$S_{obs}^{CoPol}(v_i, h_j) = S_{signal}^{CoPol}(v_i, h_j) + n(v_i, h_j) \tag{3a}$$

and

$$S_{obs}^{XPol}(v_i, h_j) = S_{signal}^{XPol}(v_i, h_j) + n(v_i, h_j) . \tag{3b}$$

The signal powers are a combination of insect signal power $S_{insect}^{CoPol/XPol}(v_i, h_j)$ [Watts] and hydrometeor signal power $S_{hydro}^{CoPol/XPol}(v_i, h_j)$ [Watts] for both polarizations. This can be expressed as

$$S_{signal}^{CoPol}(v_i, h_j) = S_{insect}^{CoPol}(v_i, h_j) + S_{hydro}^{CoPol}(v_i, h_j) \text{ and} \tag{4a}$$

$$S_{signal}^{XPol}(v_i, h_j) = S_{insect}^{XPol}(v_i, h_j) + S_{hydro}^{XPol}(v_i, h_j) . \tag{4b}$$

The goal of the CoPol and LDR insect-hydrometeor detection algorithms is to classify insect and hydrometeor scattering contributions at each $(v_i, h_j)$ location. Insects and hydrometeors do occur in the same range gate and sometimes at the same velocity (as will be seen in Figs. 6, 8, 9, and 11). These simultaneous insect and hydrometeor classifications will be mitigated by temporal quality control filtering.

The observed KAZR CoPol and XPol spectra profiles (Fig. 3) are the inputs to the insect-hydrometeor algorithms, with the processing steps for both algorithms outlined in Fig. 5. The methodology consists of two parallel algorithms. The *CoPol Texture Algorithm* classifies insects and hydrometeors based on the CoPol spectra texture, with the understanding that scattering from insects produces more spectrum variability than cloud droplet or raindrop distributions. The *LDR algorithm* classifies insects and hydrometeors based on the understanding that asymmetric insects produce larger LDR magnitudes than spherical raindrops (when

viewed from the bottom), and that the non-precipitating liquid cloud droplets should not produce any signal in the KAZR XPol channel for single scattering processes. Both algorithms produce insect-hydrometeor membership classes for every region of the spectra profile. The membership classes are combined and then reduced to binary insect and hydrometeor masks that have

affirmative values for insect or hydrometeor scattering at each range gate. After processing individual spectra profiles, two time-height continuity quality control (QC) filters are applied to the binary hydrometeor masks to remove outliers. This is based on the understanding that clouds and precipitation are persistent over 10's of seconds and 10's of meters. Details of each algorithm module are described in the following sections.

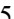

**Figure 5: Retrieval logical flow diagram**

### 4.1 CoPol Texture Algorithm Branch

This section describes the *CoPol Texture Algorithm* by applying the processing steps (Boxes 1-4 of Fig. 5) to the observed spectra

10   profile shown in Fig. 3a. An objective noise threshold $n_{HS}(h_j)$ is estimated from the CoPol spectra at each height (Hildebrand and Sekhon, 1974; Carter et al., 1995). The CoPol spectra with magnitudes greater than $n_{HS}(h_j)$ are defined as signal power (see equation 3). The CoPol signal power for the boundary layer spectra in Fig. 3a is shown in Fig. 6a. As discussed before, insect scattering produces delta functions in the power spectra that are broadened in the velocity domain because of hardware limitations (e.g., antenna beamwidth) and signal processing techniques (e.g., FFT processing). A texture parameter $T^{dB}(v_i, h_j)$ [dB] (Box 2

15   of Fig. 5) captures delta function variability in the CoPol power spectra, and is defined as

$$T^{dB}(v_i, h_j) = max\left[\left|S^{CoPol}_{signal,dB}(v_i) - S^{CoPol}_{signal,dB}(v_{i-1})\right|, \left|S^{CoPol}_{signal,dB}(v_i) - S^{CoPol}_{signal,dB}(v_{i+1})\right|\right] \tag{5}$$

where $max[a, b]$ selects the larger magnitude value between estimates $a$ or $b$. To capture both positive and negative changes equally, $T^{dB}(v_i, h_j)$ uses the absolute magnitude, then selects the largest difference between the neighbors (i.e., $v_{i-1}$ or $v_{i+1}$). Figure 6b shows the texture $T^{dB}(v_i, h_j)$ for the CoPol power spectra shown in Fig. 6a. Note that the small magnitude texture values in the upper heights are due to cloud droplet scattering and larger magnitude texture values in the lower heights are caused by

insect scattering. Several features make texture $T^{dB}(v_i, h_j)$ well suited for identifying insect produced delta function variability. First, the texture $T^{dB}(v_i, h_j)$ is calculated using signal powers expressed in decibel units. Thus, the power difference between neighbors in decibel units is the same as a power ratio, or a percent change, when the power is expressed in linear units (e.g., $10log\left[\frac{A}{B}\right] = 10log[A] - 10log[B]$). This implies that fluctuations expressed in decibel units are independent of magnitude, which allows for comparisons of low magnitude cloud observations with larger magnitude insect observations as shown in Fig. 3. Second,

a narrow KAZR antenna beamwidth allows the difference between nearest neighbors (i.e., $v_i$ and $v_{i\pm1}$) to quantify delta functions. Note that depending on radar hardware and operating parameters, the insect peak may be broader than these observations, and power differences using further neighbors may be necessary in order to identify delta functions (e.g., between $v_i$ and $v_{i\pm2}$).

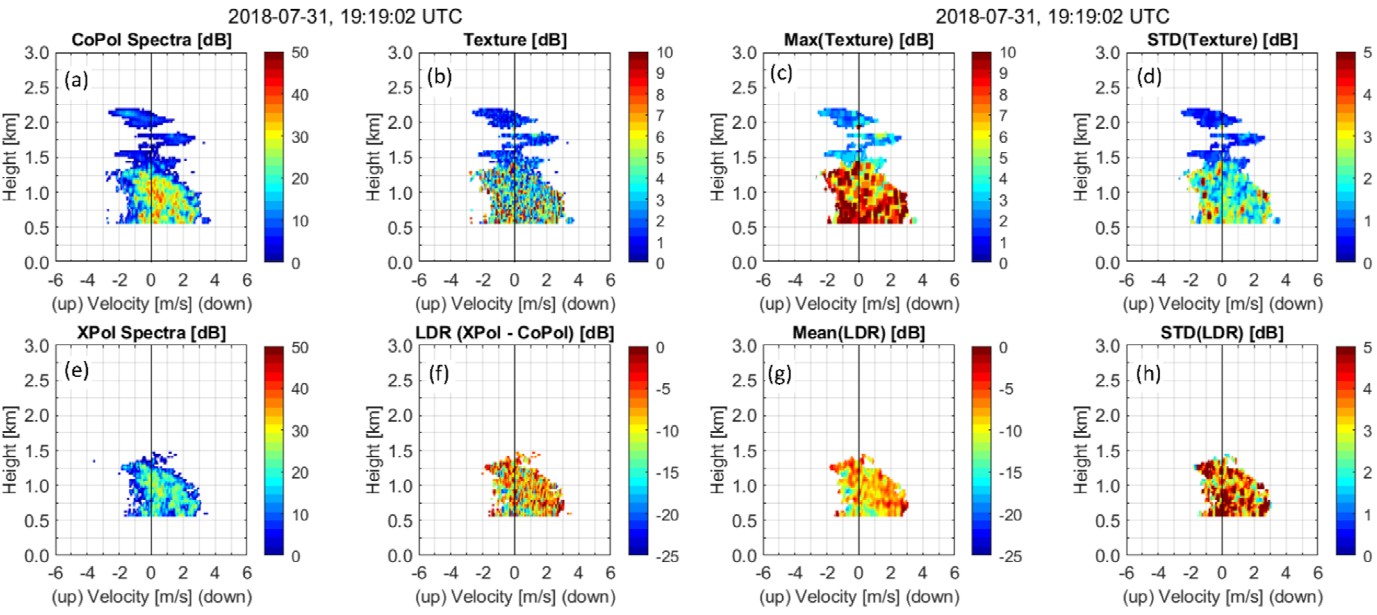

**Figure 6: Spectra profile measurements and calculations from profile at 19:19:02 UTC on 31-July-2018. (a) CoPol spectra. (b) CoPol**
**Texture. (c) Max(Texture). (d) STD(Texture). (e) XPol spectra. (f) LDR spectra. (g) Mean(LDR). (h) STD(LDR). All measurements and calculations are in units of dB.**

With a goal of identifying regions of insect and hydrometeor scattering, a small velocity-height window is moved throughout the $T^{dB}(v_i, h_j)$ domain and texture statistics are calculated for each small region. For this KAZR dataset, a velocity-height window of five (5) velocity bins (total width of 0.186 m s$^{-1}$) and three (3) range gates (total depth of 90 m) was large enough

to capture regional texture variability. For each small region, maximum texture $T^{dB}_{max} = max[T^{dB}(v_{i\pm2}, h_{j\pm1})]$ and standard deviation $T^{dB}_{STD} = STD[T^{dB}(v_{i\pm2}, h_{j\pm1})]$ are estimated and assigned to the location $(v_i, h_j)$. Figures 6c and 6d show the regional maximum and standard deviation for the texture shown in Fig. 6b. Note that both quantities are larger at lower altitudes where insect scattering dominates compared to higher altitudes that are dominated by cloud droplet scattering. Interestingly, enhancements in both max texture and STD texture are visible near 1.8 and 2 km indicating that insect scattering is occurring with

close proximity to cloud scattering regions.

With an objective of separating insect and cloud scattering regions based on CoPol texture statistics, Fig. 7a, 7b, and 7c shows 1-dimensional (1D) and 2-dimensional (2D) probability distribution functions (PDFs) of $T_{STD}^{dB} = STD[T^{dB}]$ and $T_{max}^{dB} = max[T^{dB}]$ for all profiles in hour 19 of 31-July-2018 and all spectral regions that do not have a LDR measurement. The spectral regions with a LDR measurement are shown in Fig. 7d, 7e, and 7f. The color coding in the 2D plot represents the percent occurrence relative to the cell with maximum number of observations. The 1D PDFs produced from the observations are shown with black curves in Figs. 7a and 7c using 953,136 samples, each representing a small spectral region, distributed into two populations. The peak with smaller $STD[T^{dB}]$ and smaller $max[T^{dB}]$ is due to cloud particle scattering. The peak with larger texture attributes is caused by insect scattering. The contour lines in Fig. 7b represent 90%, 75%, 63% and 50% occurrence of 2D Gaussian functions estimated for both populations. The red lines in Figs. 7a and 7c are 1D Gaussian function fits to the observations. Better fits were obtained using Generalized Gaussian functions that accounted for skewness in the observed distributions. However, these better fits did not yield better classifications, as better classifications depend on the samples between the two peaks and not on the outer tails of the distributions that determined the distribution skewness.

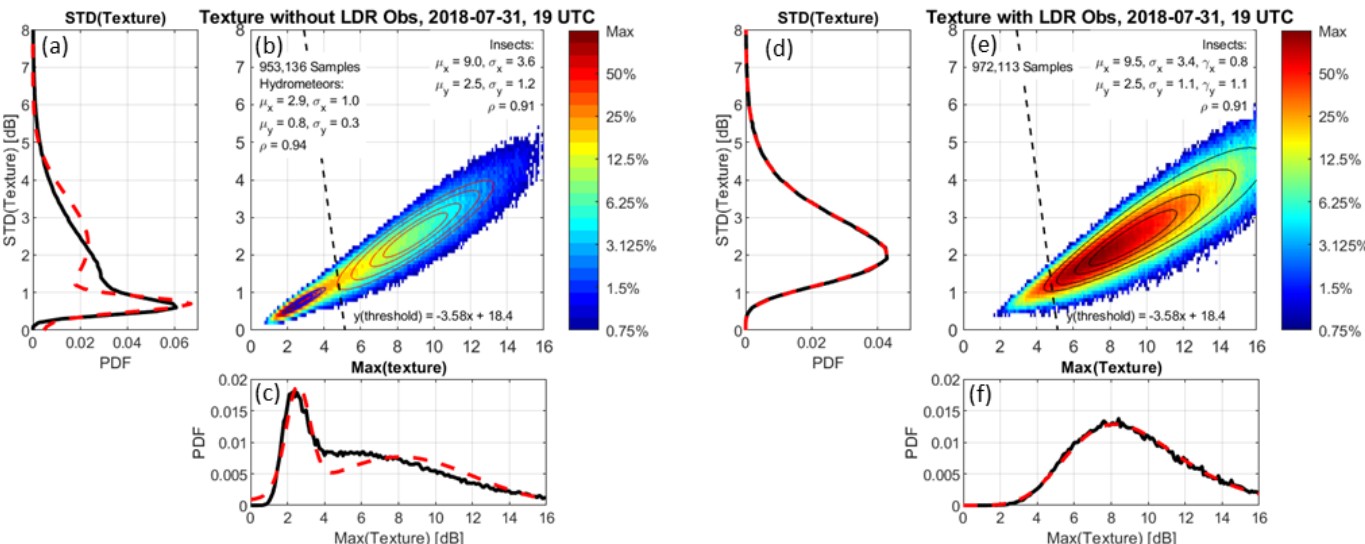

**Figure 7: 1D and 2D distributions of texture statistics from hour 19 UTC on 31-July-2018. Panels (a)-(c) are 953,136 spectra regions without LDR detected and panels (d)-(f) are 972,113 spectra regions with LDR detected. (a) 1D PDF of STD(Texture). Black line is observations and red dashed line is fit to two Gaussian distributions. (b) Colors are observed 2D distribution of STD(Texture) vs. Max(Texture). Colors represent drop from pixel with most occurrences. Blue and red contours are 2D Gaussian fits to hydrometeors (blue) and insects (red). Contours represent 90%, 75%, 63%, and 50% occurrence. Gaussian fit parameters are displayed in panel. Threshold between hydrometeor and insect indicated by dashed black line. (c) 1D PDF of Max(Texture). Black line is observations and red line is fit to two Gaussian distributions. (d) Similar to (a) except for spectra regions with detected LDR. (e) Similar to (b) except there is only one distribution caused by insect scattering. Contours are 2D General Gaussian fits. (f) Similar to (c) except red curve is fit to one General Gaussian distribution.**

The observations with LDR in Fig. 7d, e, and f show only one distribution corresponding to insect scattering. The functional fits are Generalized Gaussian distributions and capture skewness in the distributions. Note the similarities between the fitted parameters for the insect populations with and without LDR measurements. Both distributions have similar means and standard deviations (i.e., near 10 dB mean and 2.7 dB standard deviations). Also note that the insect distribution in Fig. 7e extends toward the origin and overlaps with the cloud population shown in Fig. 7b. This overlap causes difficulty in using a simple threshold to classify hydrometeor from insect observations. This difficulty was noticed in Luke et al. (2008). One way to improve the classification is to use a threshold that is orthogonal to the observed distributions. The insect and hydrometeor 2D Gaussian functional fits shown in Fig. 7b and 7e have correlation coefficients greater than 0.9 and indicate the distributions are close to a 1-

to-1 slope. After creating a line between the hydrometeor and insect distributions, an orthogonal threshold can be constructed. Figure 7b and 7e show the orthogonal threshold developed by analyzing many hydrometeor and insect observations from 2018 and 2019 (see Appendix A for details). The analysis presented in Appendix A suggests that the orthogonal threshold has a true positive rate of about 90% for both hydrometeor and insect scattering observations. Due to the distribution overlap, a single

threshold methodology will not reach 100% true positive rate and additional classification or filtering will be necessary. One way to improve the classifications due to distribution overlap or inaccurate thresholds is to apply continuity filters to remove random or ephemeral samples due to misclassifications as discussed in Section 4.4. Applying the orthogonal CoPol texture threshold to the example profile from 19:19:02 UTC, Fig. 8a shows the insect (blue shading) and hydrometeor (red shading) texture membership classes. Also in Fig. 8 are the LDR insect-hydrometeor classes; the combined classes; and the profile mask; all of which are

discussed in the next section.

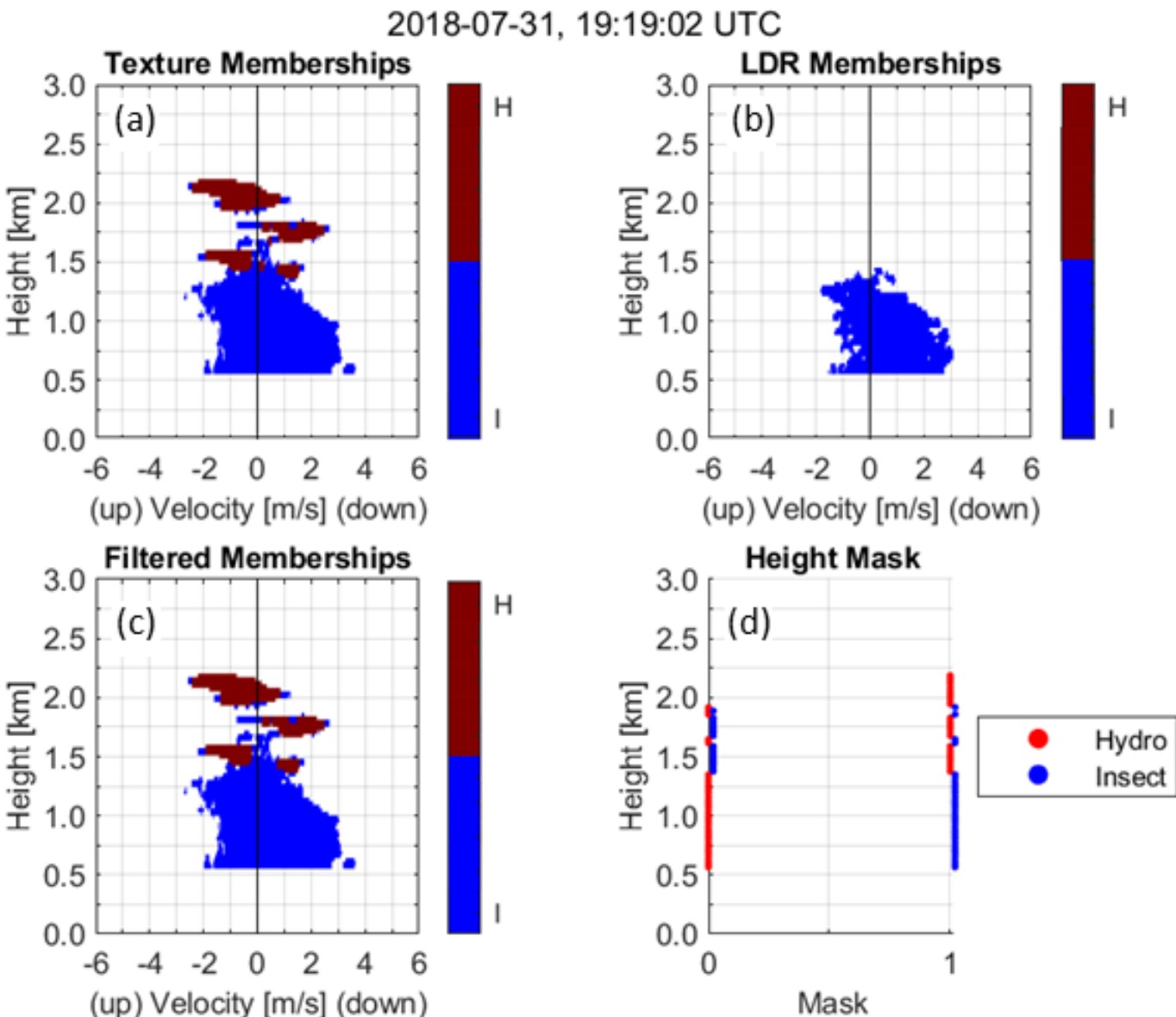

**Figure 8: Spectral memberships and binary mask for profile at 19:19:02 UTC on 31-July-2018. Red color indicates hydrometeor membership and blue color represents insect membership (a) Texture algorithm spectral membership. (b) LDR algorithm spectral**
**membership. (c) Filtered spectral membership. (d) Binary hydrometeor and insect mask.**

## 4.2 LDR Algorithm Branch

This section describes the processing steps of the *LDR Algorithm* (Boxes 5-8 of Fig. 5). In Box 5, an objective noise threshold $n_{HS}(h_j)$ is estimated from the XPol spectra at each height (Hildebrand and Sekhon 1974; Carter et al. 1995). The XPol spectra with magnitudes greater than $n_{HS}(h_j)$ are defined as signal power (see equation 3). Box 6 calculates the linear depolarization ratio spectra using equation (1). The CoPol and XPol spectra profiles at 04:48:17 UTC from the precipitation event on 4-April-2019 shown in Fig. 4 are shown in Fig. 9. The top row of Fig. 9 (Fig. 9a-d) shows CoPol observations and CoPol texture statistics used in the CoPol texture algorithm. Figures 9e and 9f show XPol and LDR spectra profiles. To estimate regional scattering properties, the same 5x3 velocity-height window used in the texture algorithm is used to calculate regional LDR statistics throughout the $S_{dB}^{LDR}(v_i, h_j)$ spectra profile (Box 7 of Fig. 5). Figures 9g and 9h show the $mean[S_{dB}^{LDR}(v_i, h_j)]$ and $STD[S_{dB}^{LDR}(v_i, h_j)]$ estimates and suggest that insects are present below 1 km with near zero vertical velocity and falling hydrometeors are present above 3 km. The insects are deduced by $mean[S_{dB}^{LDR}(v_i, h_j)]$ between -10 and -5 dB and the falling hydrometeors by $mean[S_{dB}^{LDR}(v_i, h_j)]$ less than -20 dB. These inferences are supported by the CoPol texture statistics (Figs. 9c and 9d) with insects having large $max[T^{dB}(v_{i\pm2}, h_{j\pm1})]$ near zero vertical velocities below 1 km and smaller values elsewhere. As with the warm shallow cumulus cloud event shown in Fig. 6, there are more CoPol observations (Fig. 6a-d) than LDR measurements (Fig. 6e-h) below 1.5 km.

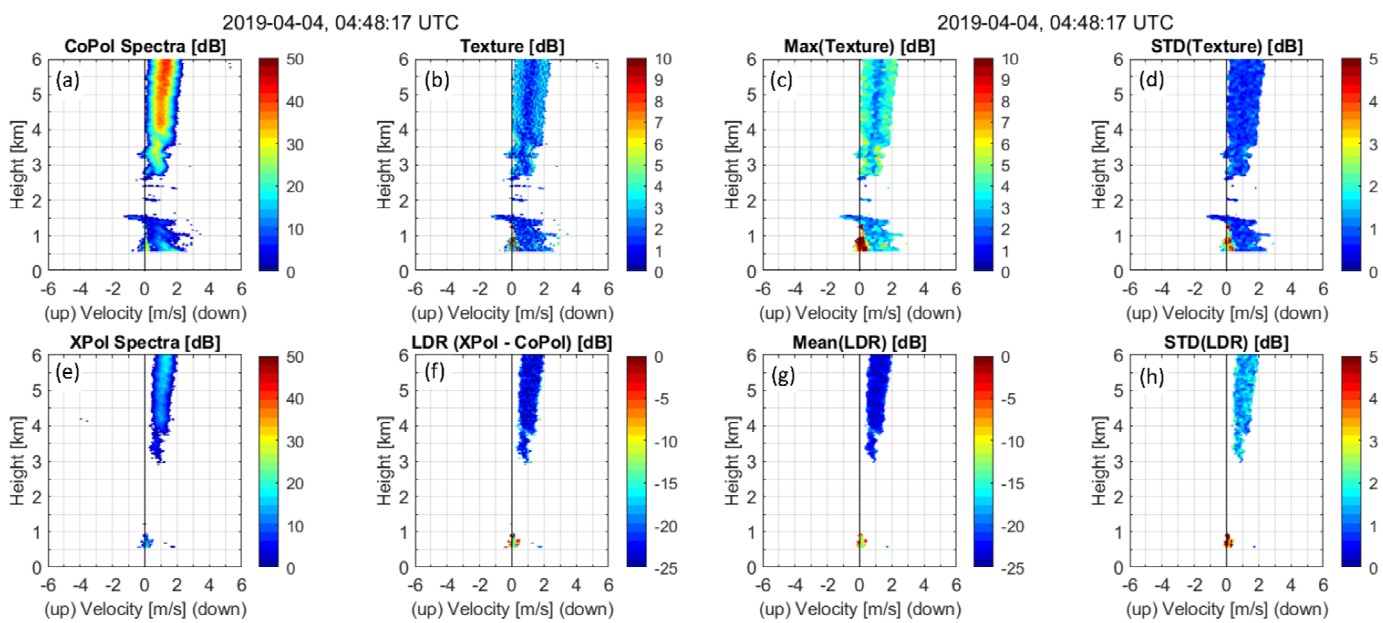

**Figure 9: Same as Fig. 6 except for profile at 04:48:17 UTC on 4-April-2019.**

With an objective of separating insect and hydrometeor scattering regions based on LDR statistics, Fig. 10 shows 2D and 1D PDFs of the LDR statistics estimated for all observations below 1.5 km (to avoid too many hydrometeor observations that would prevent any insects from appearing in Fig. 10) for hour 04 on 04-April-2019. Figure 10 contains over 1 million LDR statistic samples each calculated over a separate 5x3 spectral region. The distribution near $mean[S_{dB}^{LDR}(v_i, h_j)] = -8$ dB is due to insect scattering and the distribution near $mean[S_{dB}^{LDR}(v_i, h_j)] = -20$ dB is due to hydrometeor scattering. A threshold of $mean[S_{dB}^{LDR}]_{threshold} = -15$ dB clearly separates the two distributions and is indicated with a dashed line in Fig. 10b, which is consistent with estimates from Matrosov (1991) and Reinking et al. (1997).

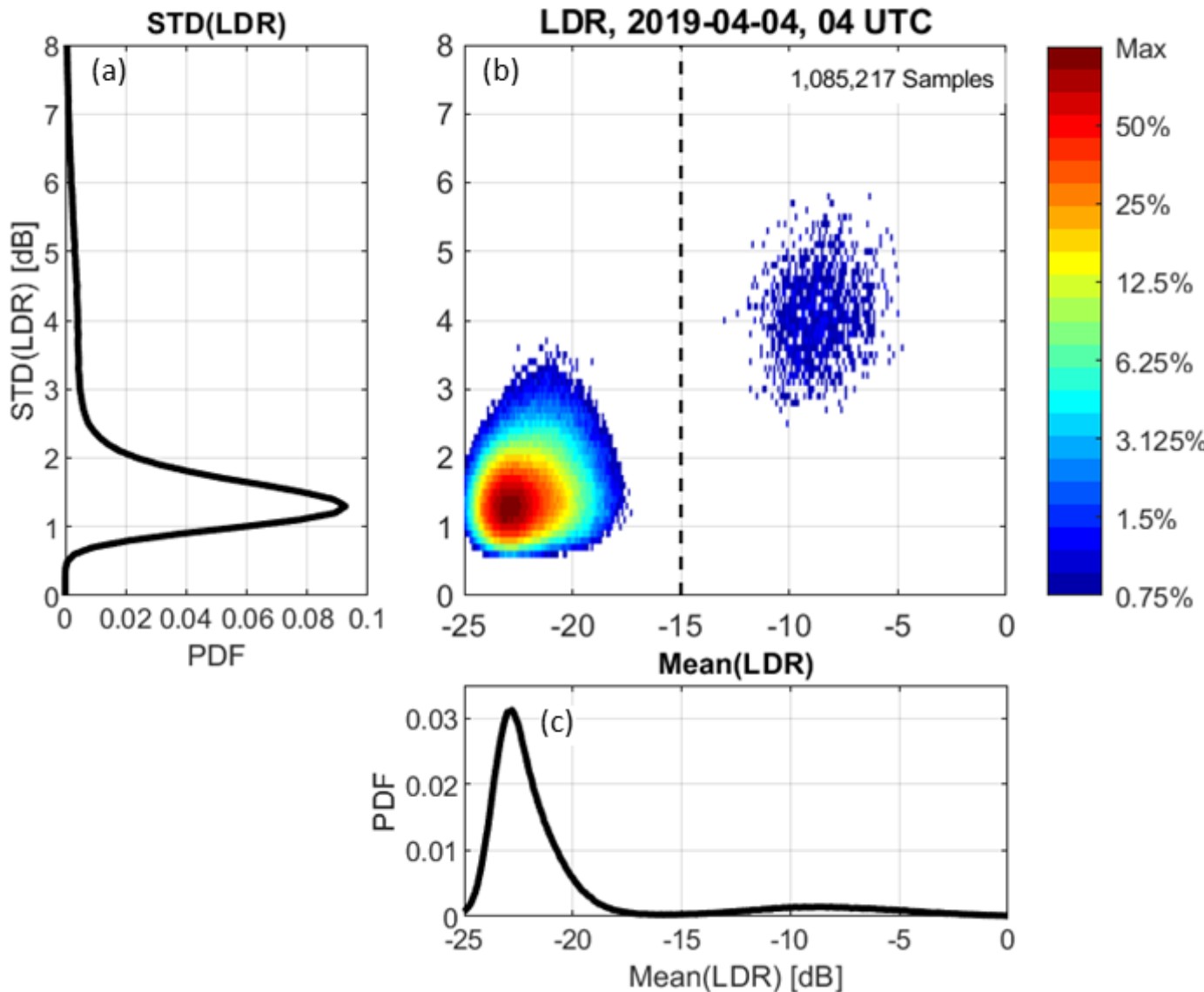

**Figure 10: Similar to Fig. 7 except for hour 4 UTC on 4-April-2019 and for STD(LDR) and mean(LDR) statistics. There are 1,085,217 samples collected below 1.5 km height.**

Figures 11a and 11b show the CoPol texture and LDR membership classes for this spectra profile. Blue shading indicates insect scattering and red shading indicates hydrometeor scattering. Note that the texture algorithm identifies both insect and hydrometeor scattering below 1.5 km while the LDR algorithm only detects a few insects at these lower range gates. Both algorithms identify hydrometeor scattering above about 3 km.

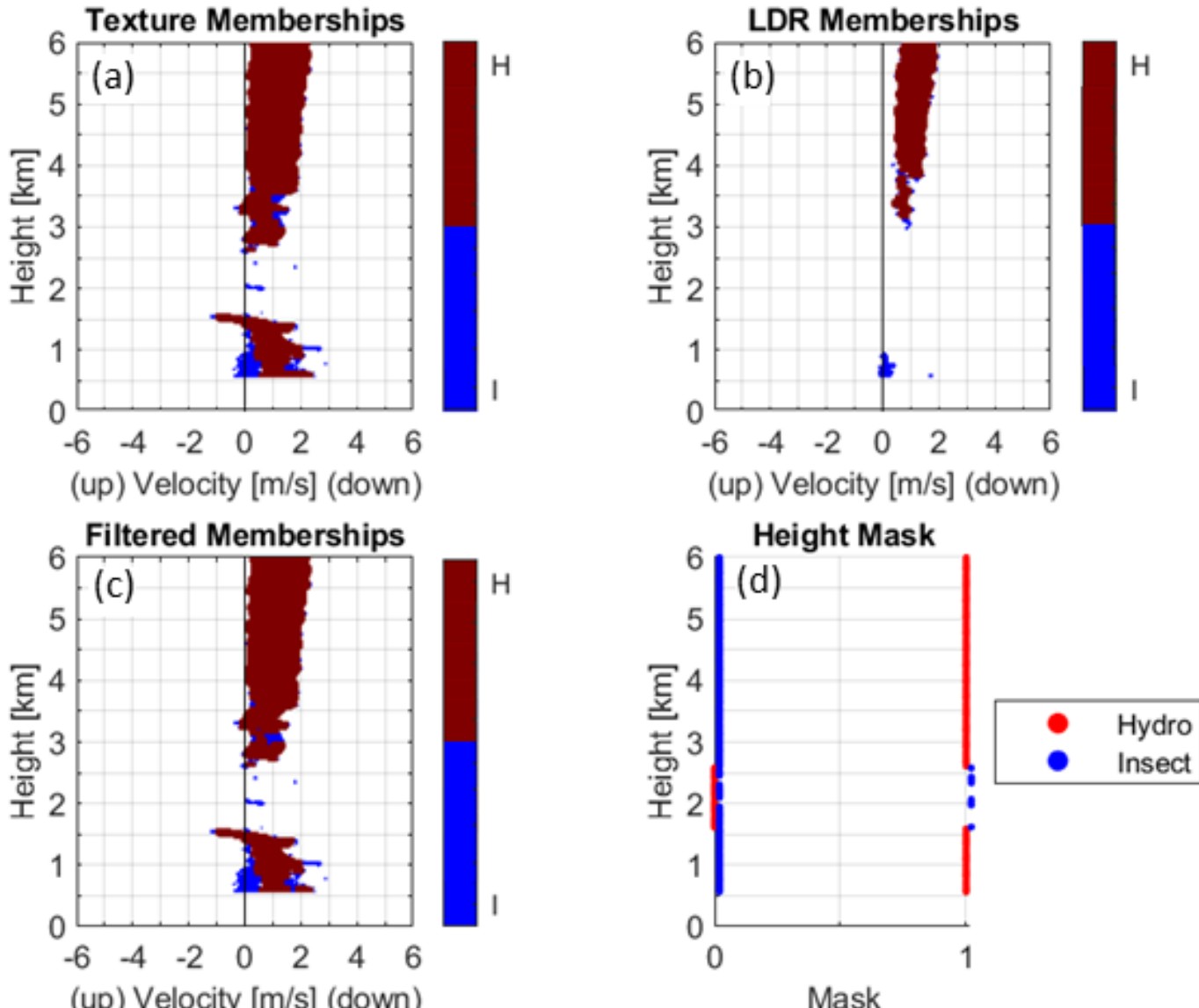

**Figure 11: Same as Fig. 8 except for profile at 04:48:17 UTC on 4-April-2019.**

### 4.3 Combining Co-Pol Texture and LDR Algorithm Classifications

After performing the CoPol texture and LDR algorithms, the binary insect and hydrometeor spectral classifications from both
5   algorithms are combined and then filtered (e.g., see Figs. 8a and 8b, and Figs. 11a and 11b). Initially, the combined spectral classification is the texture classification because the LDR classification will always have fewer valid observations than the CoPol observations. To incorporate the LDR classification, the combined classification is changed only if the LDR algorithm produced a hydrometeor class when the texture classification was set to insect class. This logic places more emphasis on identifying hydrometeors than insects.

10       One of the physical attributes of hydrometeor scattering is that the Doppler velocity spectra span multiple continuous velocity bins and over several range gates. Accordingly, isolated hydrometeor pixels in the combined spectral classification are removed by requiring at least 7 continuous hydrometeor pixels in the velocity dimension. All hydrometeor pixels not satisfying this constraint are set to the insect scattering class. The filtered memberships for the two example profiles are shown in Figs. 8c

for the warm liquid cloud event on 31-July-2018 and Fig. 11c for the precipitation event on 4-April-2019. The red and blue shading corresponds to hydrometeor and insect scattering classes, respectively.

The final processing step is to reduce the filtered membership classes into binary masks indicating the presence of insect or hydrometeor scattering at each range gate (Box 10 of Fig. 5). The insect and hydrometeor masks are set to unity if that filtered membership class exists for that range gate $h_j$. In the case when both insect and hydrometeor scattering are detected at the same range gate, the hydrometeor mask is set to unity and the insect mask is set to zero. This logic places more emphasis on identifying robust hydrometeor masks and defining masks resulting from either insect or hydrometeor scattering at each range gate. Figures 8d and 11d show the binary insect and hydrometeor masks for the two example profiles. Both masks are saved in output data files and have the variable names *insect_mask_raw* and *hydro_mask_raw* (Boxes 11 and 12 of Fig. 5). The suffix *raw* designates that these masks were estimated from individual profiles and without any temporal information from neighboring profiles, which is discussed in Section 4.4.

In addition to the binary insect mask, an insect activity index is generated by counting the number of insect scattering velocity bins at each height. This insect index $I_{insect}(h_j)$ is defined as

$$I_{insect}(h_j) = \sum_{i=1}^{i_{max}} C_{insect}^{filtered}(v_i, h_j) \tag{6}$$

where $C_{insect}^{filtered}(v_i, h_j)$ is the insect spectral classification and has a value of either 0 or 1. This insect index is not an estimate of the insect number concentration because the magnitude of the insect scattering is not being taken into account. The authors hypothesize that the insect index should be related to insect number density, as the breadth of insect velocities should increase as the number of insects increases. The insect index is available in the output data files with the variable name *insect_index_raw*.

## 4.4  Quality Control (QC) filtering of the Cloud Profile Mask

Figure 12 shows the time-height cross-section of observed CoPol KAZR reflectivity (Fig. 12a), the raw hydrometeor mask (Fig. 12b), a time-height filtered hydrometeor mask (Fig. 12c), and the insect index (Fig. 12d) for hour 19 on 31-July-2018. This is the same event shown in Figs. 1 and 2, except with the vertical axis limited to 3 km height. The ceilometer cloud base height is shown in each panel with black dots. The blue and red plus symbols are cloud top and base determined from the COGS stereo camera system, which is discussed in more detail in Section 5. The hydrometeor mask in Fig. 12b is the raw mask produced from each spectra profile. These raw hydrometeor masks contain random misclassified pixels of hydrometeors below the ceilometer cloud base height. Most of these false positive hydrometeor mask pixels are removed by sequentially applying two time-height quality control filters.

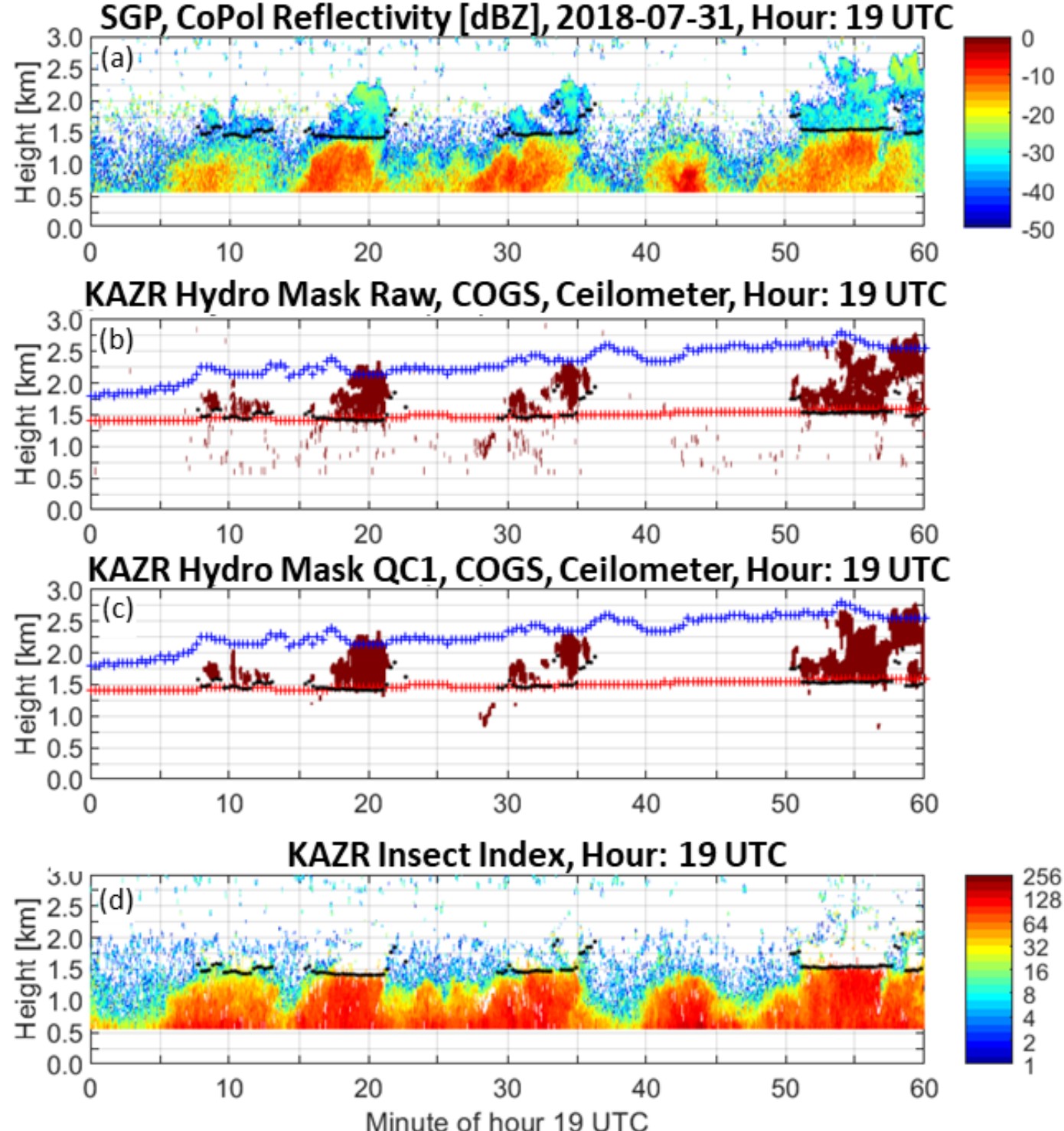

**Figure 12: Measurements and retrievals for hour 19 UTC on 31-July-2018. (a) CoPol reflectivity [dB]. (b) Retrieved hydrometeor raw mask (red shading). COGS-derived 6-km x 6-km domain average cloud base (red symbols) and cloud top (blue symbols). (c) Same as panel (b) expect for retrieved hydrometeor QC1 mask. (d) Retrieved insect activity index. Black symbols are ceilometer-derived cloud base.**

The first quality control filter, named QC1 (shown in Fig. 12c), removes temporal outliers by applying a 3-member temporal continuity filter, which retains all three values if three consecutive values are present. The QC1 filter also inserts up to three consecutive hydrometeor mask pixels in vertical profiles to fill small gaps in the raw hydrometeor mask. The second quality control filter, named QC2 (not shown), applies a low-pass filter to the QC1 filtered mask by moving a 3x3 time-height

(approximately 12 s by 90 m) continuity constraint throughout the domain to robustly identify hydrometeors that are persistent in time and height. Both the QC1 and QC2 filtered hydrometeor masks are available in the output data files with variable names *hydro_mask_qc1* and *hydro_mask_qc2*. Figure 12d shows the insect index and estimates the number of velocity bins in the spectra that contained insect scattering. The color scale is logarithmic with maximum value 256 representing the number of velocity bins in the spectra. The QC1 hydrometeor mask is plotted for the 4-April-2019 precipitation event in Fig. 4d. The mask identifies the shallow clouds near 1.5 km from about 2-to-13 minutes and the precipitating anvil at the beginning of the hour between 3-to-4 km that descends to the lowest range gate just after minute 20. The hydrometeor mask below 1.5 km starting at about minute 21 and continues until the end of the hour except for a shallow gap between minutes 50-to-55 is due to precipitation at these lower heights as indicated in Fig. 11c. There is strong agreement between the ceilometer cloud base height estimates and the hydrometeor mask before minute 20. After this time, the hydrometeor mask identifies raindrops, while the ceilometer is identifying cloud base. COGS measurements are unavailable for comparison purposes during this event because COGS is an optical system requiring daylight.

## 5.   Comparing Cloud Mask with Independent Measurements

Figures 4 and 12 show significant agreement between ceilometer cloud base estimates and retrieved QC1 hydrometeor masks. Figure 12 also shows agreement between COGS cloud base and top estimates with the QC1 hydrometeor mask. In comparing the three products, the KAZR hydrometeor masks and ceilometer cloud base estimates appear as discrete cloud events. Conversely, the COGS estimates appear continuous in time, as if COGS is detecting a persistent cloud layer. This difference is because KAZR and ceilometer are 'soda-straw' observations and COGS is a 6-km x 6-km domain-averaged product produced from three pairs of stereo cameras positioned around the radar and ceilometer (Romps and Öktem, 2018). Figure 12c shows that when the radar and ceilometer both detect clouds, COGS also had a similar cloud base height estimate. The ceilometer and radar cloud bases also showed consistency even at the cloud edges (see near minute 35). Regarding cloud top estimates, COGS estimates are higher than the radar because COGS is a domain average. The online Supplemental Material section contains images of QC1 hydrometeor mask, ceilometer, and COGS retrievals for forty-seven (47) days corresponding to 2018 and 2019 LASSO shallow cloud events (LASSO, 2020). The COGS product is available only for shallow cumuliform clouds and only during daylight hours.

Figure 13 compares hydrometeor mask QC1 column bottoms with ceilometer cloud bases for the 47 LASSO days. The hydrometeor mask QC1 columns were at least 90 m thick (i.e., 3 consecutive range gates). Using the same format as Figs. 7 and 10, Fig. 13b shows the 2D distribution of height differences with the line graphs showing 1D PDFs. Over 70% of the 12,141 simultaneous profiles had height differences within +/-100 m, which represents +/-3 thirty-meter radar range gates. There is a small skewness to the height difference PDF (Fig. 13a) that is consistent with the ceilometer detecting clouds before the radar detects hydrometeors. Also, during the few precipitation events, the hydrometeor mask bottom was significantly lower than the ceilometer cloud base as the hydrometeor mask detects falling raindrops far below the ceilometer detected cloud base.

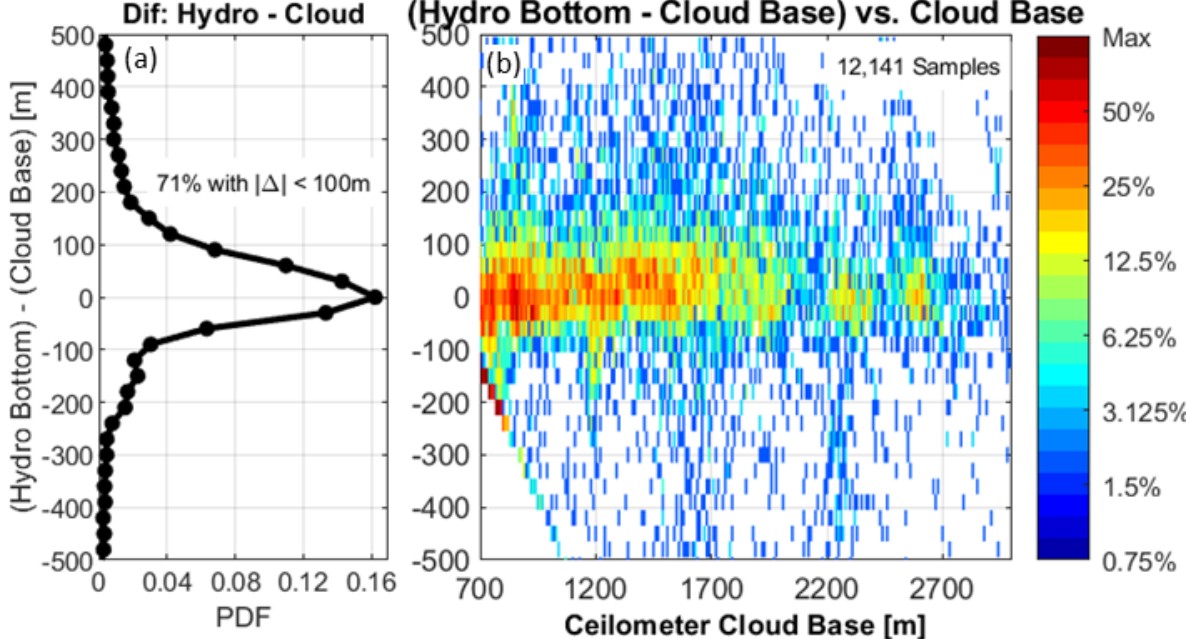

**Figure 13: Difference in hydrometeor mask QC1 column bottom and ceilometer cloud base height using 47 days at SGP during 2018 and 2019. There were 12,141 profiles with simultaneous hydrometeor mask QC1 and ceilometer cloud bases. (a) 1D PDF of height difference defined as (Hydrometeor mask column bottom) – (Ceilometer cloud base) [m] with 30-m resolution corresponding to radar range resolution. (b) Colors are 2D distributions of height difference vs. ceilometer cloud base. Colors represent drop from pixel with most occurrences. Artifact at negative height differences and low ceilometer cloud base is due to radar first range gate at 570 m.**

## 6.  Conclusions

In addition to detecting cloud particles, vertically pointing cloud radars are sensitive enough to detect individual insects. If insect contamination is not identified and removed, then radar derived cloud properties will be incorrect and will not help with validating cloud resolving models or climate simulations. This study used polarimetric radar observations to develop two insect-hydrometeor detection algorithms. The two algorithms use different radar scattering principles to identify small velocity-height regions in the Doppler velocity power spectra profile as resulting from either insect or hydrometeor scattering. The results of both algorithms are combined and then filtered to produce single value insect and hydrometeor masks at each range gate. The backscattered power from hydrometeors and insects is larger in the CoPol channel than the XPol channel leading to negative LDR values. This difference in sensitivity leads to this study finding that KAZR XPol spectra observations observed fewer insects than KAZR CoPol observations. This implies that using just a polarimetric signal processing method to identify insects will not identify all insect clutter affecting CoPol observations and that insect clutter mitigation methods must use CoPol observations to identify all insect clutter in the CoPol channel.

One algorithm uses the texture of CoPol Doppler velocity power spectra to identify small velocity-height regions of spectra attributed to insect or hydrometeor scattering. Since insects are individual point targets, their radar power return is confined to narrow intervals of Doppler velocity and range gates, on the order of 1-to-3 velocity bins (0.04 to 0.12 m s$^{-1}$) and 1-to-3 range gates (30 to 90 m). In contrast, cloud particles and raindrops occur in distributions that extend over many velocity bins and several range gates, on the order of 5-to-150 velocity bins (0.2 to 7 m s$^{-1}$) and 3-to-150 range gates (90 to 4500 m). The CoPol and XPol

Doppler velocity power spectra from insect scattering have large variability, or texture, while scattering from cloud particles and raindrops produce smoother, less variable, spectra. The CoPol texture algorithm uses the texture information to identify small regions of insect and hydrometeor scattering. The CoPol texture algorithm can be applied to any cloud radar system collecting Doppler velocity power spectra and does not require a cross-polarization channel.

The other algorithm uses the linear depolarization ratio (LDR) at each point in the Doppler velocity power spectra to identify regions of scattering due to spherical raindrops, asymmetric ice particles, or asymmetric insects. Unlike previous studies, this work uses the LDR at each spectra bin. After identifying small velocity-height regions of insect and hydrometeor scattering in both algorithms, the spectra classifications are combined and then filtered to account for continuity in the Doppler velocity and vertical range dimensions. The filtered spectra classifications are reduced to binary affirmative insect and hydrometeor masks with

a single value at each range gate. An insect activity index is estimated at each range gate by counting the number Doppler velocity spectra bins with insect scattering. Future studies will use insect activity and vertical air motion estimates to explore whether insects are passive tracers or actively propelling themselves through the atmosphere. Often, insects occur at the same height as clouds and during the onset of precipitation. While these are interesting phenomena, the focus of this work is producing robust hydrometeor masks to help identify cloud boundaries, which can be used, for example, to study the evolution of shallow cumulus

clouds in the planetary boundary layer (Gustafson et al., 2017). Using over 12,000 simultaneous ceilometer and radar profiles, it was found that over 70% of the hydrometeor mask column bottoms were within +/-100 m of the ceilometer cloud base (i.e., +/-3 thirty-meter radar range gates). The hydrometeor mask column bottom was slightly higher than the ceilometer cloud base. This is to be expected, as the ceilometer detects cloud particles at lower heights than the radar detecting hydrometeors within the cloud.

The online Supplemental Material includes sample images of observed KAZR reflectivity, retrieved hydrometeor masks,

and verification observations from ceilometer and COGS. The processing described herein was applied to KAZR observations for April-October in 2018 and 2019 summer seasons at the Southern Great Plains (SGP) facility. The insect and hydrometeor masks for these months are available online on the DOE ARM Archive (Williams, 2021).

**Appendix A**

This appendix uses a large, hand edited, dataset to first explore the representativeness of spectral region texture distributions and then to develop an orthogonal threshold to classify hydrometeors from insects in the 2D texture distributions. Over 75 hours of KAZR observations from 2018 and 2019 at SGP were manually inspected to contain only hydrometeors or insect scattering observations. Figure A1 illustrates examples of manual classified hydrometeor and insect boundaries shown as gray shaded areas superimposed onto KAZR CoPol Reflectivity. The top panel (Fig. A1a) shows hydrometeor boundaries from 4 to 15 km for hour 12 UTC on 3-August-2019. There are no insects detected above 4 km and all observations are due to hydrometeor scattering. Figure A1b shows insect boundaries from the lowest resolved range gate to 3 km for hour 09 UTC on 06-June-2019. The ceilometer derived cloud base indicates clouds are detected above 3.5 km between minutes 14 and 37. In this 'truth' dataset, there were over 15 hours of hydrometeor profiles and over 60 hours of insect profiles. The manually classified dataset contained over 47 million CoPol and over 20 million LDR spectral regions with 5 velocity bins and 3 range gates (i.e., 0.186 m s$^{-1}$ by 90 m).

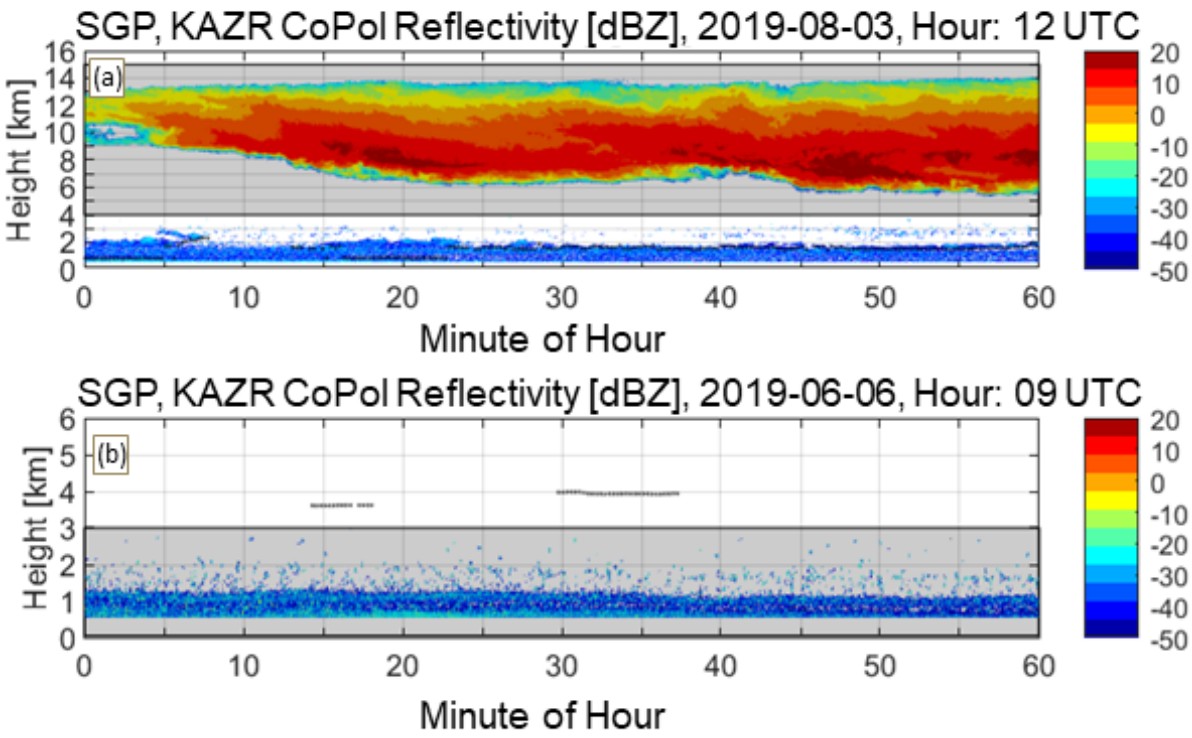

**Figure A1. Examples of manual hydrometeor and insect classification superimposed on KAZR CoPol Reflectivity at SGP. (a) Gray bounding box between 4 and 15 km contains manually identified hydrometeor scattering observations for hour 12 UTC on 3-August-2019. (b) Gray bounding box from lowest range gate to 3 km contains manually identified insect scattering observations for hour 09 UTC on 6-June-2019.**

The texture, $T^{dB}$, spectral region statistics were calculated for the hydrometeor and insect scattering truth datasets. The texture statistics of $STD[T^{dB}]$ and $\max[T^{dB}]$ are shown in Fig. A2 as 1D and 2D distributions following the format shown in Fig. 7. The hydrometeor scattering observations are shown in the left panels (Figs. A2a, A2b, and A2c) and the insect scattering observations are shown in the right panels (Figs. A2d, A2e, and A2f). The colors indicate the observed distributions and the contour lines represent 90%, 75%, 63% and 50% occurrence levels of a fitted 2D General Gaussian function. The fitted parameters shown in Fig. A2b and A2e are the mean ($\mu_x$ and $\mu_y$), standard deviation ($\sigma_x$ and $\sigma_y$), and skewness ($\gamma_x$ and $\gamma_y$) of a 2D General Gaussian function. The subscripts $x$ and $y$ correspond to the horizontal ($\max[T^{dB}]$) and vertical ($STD[T^{dB}]$) axis parameters, respectively.

Note that the correlations of the General Gaussian functional fits are estimated separately for both datasets and both estimates are greater than 0.9 indicating that the individual distributions have nearly 1-to-1 slopes.

To construct an orthogonal classification threshold to divide observations into either hydrometeor or insect scattering classes, a line is first constructed between the two distributions and then an orthogonal slope is estimated from that original line. The slope between the two distributions is estimated from the Gaussian distribution mean values using

$$m_{observations} = \frac{\left(\mu_y^{insect} - \mu_y^{hydro}\right)}{\left(\mu_x^{insect} - \mu_x^{hydro}\right)} \tag{A1}$$

where the numerator is the change in $STD[T^{dB}]$ and the denominator is the change in $\max[T^{dB}]$. The equation of the line between the two distributions is written in Fig. A2b and A2e and is shown with the solid black line. The asterisks indicate the distribution mean locations, specifically, $\left(\mu_x^{hydro}, \mu_y^{hydro}\right)$ and $\left(\mu_x^{insect}, \mu_y^{insect}\right)$.

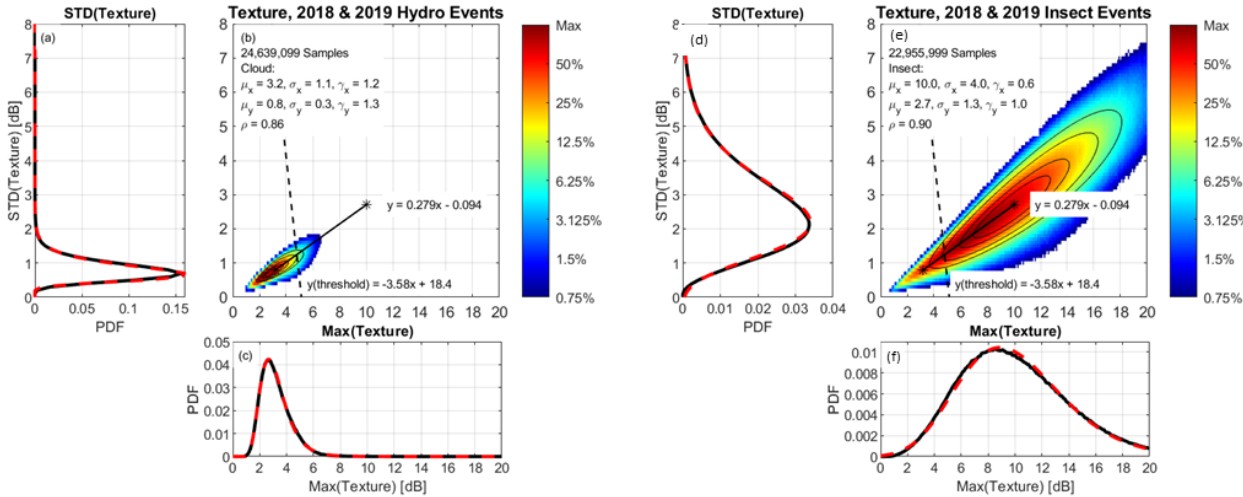

**Figure A2. Observed spectral region texture distributions using the manual classified (a)-(c) hydrometeor scattering observations and (d)-(f) insect scattering observations. The 1D and 2D distribution formats are similar to Fig. 7. The colors in (b) and (e) represent observed distributions normalized to the pixel with the most observations. The contours are the 90%, 75%, 63%, and 50% occurrences of a fitted 2D General Gaussian function. The mean, standard deviation, and skewness parameters are listed in (b) for the hydrometeor scattering distribution and in (e) for the insect scattering distribution. The solid line and equation represent a line between the hydrometeor and insect scattering distribution centers with the asterisks placed at the distribution mean values. The dashed line is the orthogonal threshold line that is the intersection of the hydrometeor and insect true positive rates (TPRs). The solid and dashed lines are orthogonal but appear non-orthogonal in the panels because of axis scaling.**

Orthogonal thresholds will have a slope given by

$$\gamma_{threshold} = -\left[\frac{1}{m_{observations}}\right] = -\left[\frac{\left(\mu_x^{insect} - \mu_x^{hydro}\right)}{\left(\mu_y^{insect} - \mu_y^{hydro}\right)}\right]. \tag{A2}$$

Using the threshold slope in (A2), many orthogonal threshold lines were constructed and the two truth datasets were classified using each threshold. The goodness of classification was determined using a receiver operating characteristic (ROC) curve with true positive rates (TPRs) estimated as the ratio of number of true positive (TP) classifications to total number of observations ($N$) using

$$TPR_{hydro} = \frac{TP_{hydro}}{N_{hydro}} \tag{A3a}$$

$$TPR_{insect} = \frac{TP_{insect}}{N_{insect}} \tag{A3b}$$

Figure A3 shows the TPRs for both datasets as the parallel thresholds moved graphically from left-to-right in Fig. A2b and A2e with increasing $\max[T^{dB}]$. Note that $TPR_{hydro}$ starts at a low value and increases as the threshold moves to larger $\max[T^{dB}]$. The

$TPR_{insect}$ has an opposite behavior. The intersection of $TPR_{insect} = TPR_{hydro}$ indicates the same true positive rate for both datasets. The intersection has a true positive rate over 0.9 and occurs when $\max[T^{dB}]$ is equal to 4.8.

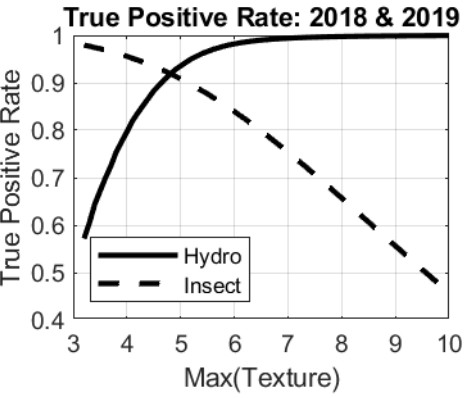

**Figure A3. True Positive Rate (TPR) for truth hydrometeor and insect scattering observations for different orthogonal thresholds. As** $\max[T^{dB}]$ **increases, the threshold moves from left-to-right up the curve** $y = 0.279x - 0.095$ **shown in Figs. A2b and A2e. The solid line is** $TPR_{hydro}$ **and the dashed line is** $TPR_{insect}$**. The intersection occurs when** $\max[T^{dB}] = 4.8$**.**

The LDR spectral region statistics were calculated for both truth datasets. The LDR statistics of $STD[LDR]$ and
mean[$LDR$] are shown in Fig. A4 as 1D and 2D distributions following the format shown in Fig. 10. Similar to Fig. A2, the hydrometeor scattering observations are shown in the left panels (Figs. A4a, A4b, and A4c) and the insect scattering observations are shown in the right panels (Figs. A4d, A4e, and A4f).

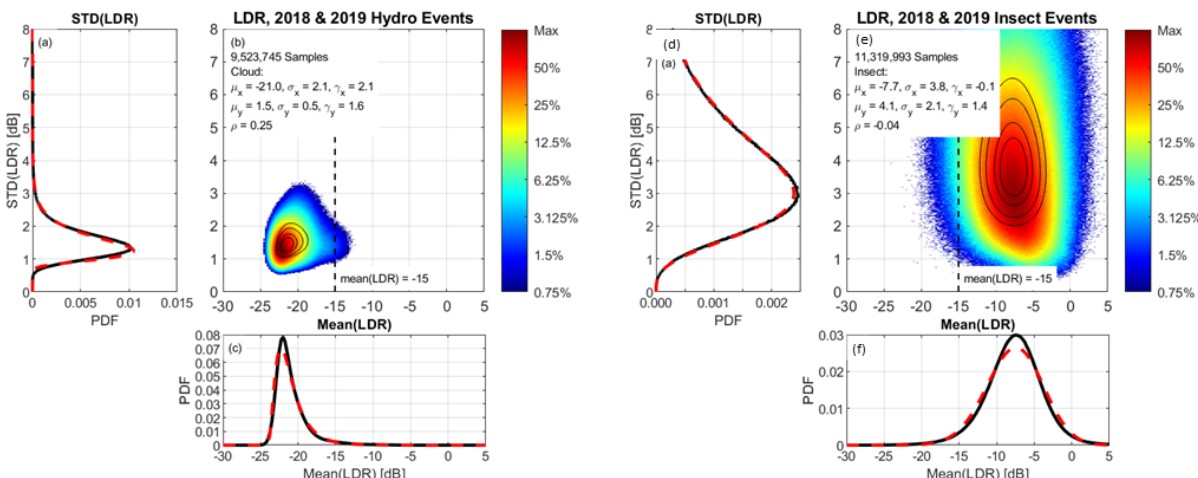

**Figure A4. Observed spectral region linear depolarization ratio (LDR) distributions using the manual classified (a)-(c) hydrometeor scattering observations and (d)-(f) insect scattering observations. The 1D and 2D distribution formats are similar to Fig. 10 and to Fig. A2.**

*Data availability*. Original raw KAZR spectra are available on the DOE ARM archive (ARM user facility, 2011a and 2011b). The algorithms described herein were applied to fourteen months (April-October 2018 and 2019) of KAZR observations from the DOE ARM Southern Great Plains (SGP) Central Facility. The produced insect and hydrometeor mask data files in netCDF format, hourly images in TIF format, and animations of individual profiles in MP4 format are available on the DOE ARM archive as a

Principle Investigator (PI) Product (Williams, 2021).

*Source code availability*. The code used to generate the insect, cloud, precipitation, and hydrometeor masks stored on the DOE ARM archive is available upon request from the lead author. With this source code, users can repeat the analysis presented in this study and develop improved insect-cloud and insect-precipitation detection algorithms for their vertically pointing radar

observations.

*Supplemental Material.* Selected images of observed KAZR reflectivity, retrieved hydrometeor masks, and verification observations from ceilometer and COGS are available in the online Supplemental Material.

*Author contribution*. CRW: insect and hydrometeor mask code development; KLJ: testing and evaluation of products and verification; SEG and JCH: Evaluation of products against collocated observations; RO and DMR: processing of COGS product.

*Competing interests*. The authors declare that they have no conflict of interest.

*Disclaimer*. The authors declare no disclaimers.

*Acknowledgements*. We recognize and appreciate the work of field technicians tasked with keeping instruments running at U.S. Department of Energy (DOE) field sites. CRW received support from the DOE Atmospheric Radiation Mission (ARM) program under Contract No. 508641. This research was supported by the DOE Office of Biological and Environmental Research as part of the ARM Climate Research Facility, an Office of Science scientific user facility. RO and DMR acknowledge support from the DOE Atmospheric System Research (ASR), an Office of Science, Office of Biological and Environmental Research program;

Lawrence Berkeley National Laboratory is operated for the DOE by the University of California under Contract No. DE-AC02-05CH11231. This paper has been authored by employees (KLJ and SEG) of Brookhaven Science Associates, LLC, under contract No. DE-SC0012704 with the U.S. Department of Energy (DOE). The publisher by accepting the paper for publication acknowledges that the United States Government retains a nonexclusive, paid-up, irrevocable, worldwide license to publish or reproduce the published form of this paper, or allow others to do so, for United States Government purposes.

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
