# Peer review of "Identifying Insects, Clouds, and Precipitation using Vertically Pointing Polarimetric Radar Doppler Velocity Spectra"

_Atmospheric Measurement Techniques, 2021_

## Referee Comment (RC2)

The manuscript titled "Identifying Insects, Clouds, and Precipitation using Vertically Pointing Polarimetric Radar Doppler Velocity Spectra" by Williams et al. describes a two-part algorithm to distinguish radar signal related to hydrometeors or insects in vertically pointing polarimetric Doppler radar measurements. Two techniques, one relying on the morphology of the co-linear (CoPol) Doppler spectrum and the other on the linear depolarization ratio (LDR) specrum, are independently applied and the resulting hydrometeor/insect masks are then combined. The performance of the method is shown by a good agreement between the hydrometeor mask and cloud base height retrieved from ceilometer, which provides an independent observation.

The authors make a good effort to illustrate the problem they are addressing (Figs. 1 and 2) as well as the reasoning behind the chosen approaches (Section 3, Figs. 3, 4, 6 and 9). Variables are clearly defined, and except a few small exceptions (see specific comments below) the description of the algorithm is detailed enough to easily follow what has been done. I also appreciate that the authors provide so many cases with the resulting hydrometeor mask in the supplemental material. The authors are addressing a common issue with cloud radar observations and although the concepts behind the described algorithm are not novel, the manuscript provides the community with a practically applicable method. My main concern with the algorithm is related to the choice of thresholds: the authors demonstrate their choice of thresholds (Figs. 7 and 10) using a sub-set of one hour data for two example cases, one case for each algorithm branch, without explaining why the specific sub-set was taken or if the thresholds obtained for the one hour example cases are generalizable to other cases. Furthermore, I find the presentation of the work lacking and needs to be improved before the manuscript can be published.

**Minor comments**

1. Page 1 Lines 15-17 (abstract) "The insect-hydrometeor discrimination method uses CoPol and XPol spectral information in two separate algorithms..."
   This sentence might be misleading because the algorithm does not utilize the XPol spectra as such, but the LDR spectra. Although strictly speaking the LDR is based on "XPol spectral information", I'd suggest to reformulate for clarity.

2. P. 1 L. 25-26. Gives the impression that the hydrometeor mask bottoms are always within +/- 100 m from ceilometer cloud base height when Fig. 13a indicates that this is the case for 71% profiles. The authors should acknowledge here that although most, not all, hydrometeor mask bottoms are within the +/- 100 m from the ceilometer base to accurately reflect the content of the paper in the abstract.

3. P. 2 L. 26-27 and Fig. 1. Information on where observations in Fig. 1 are obtained is missing.

4. P. 2 L. 27-34 and Fig. 1. The example nicely illustrates the motivation for the study. Have the authors considered also showing in Fig. 1 the reflectivity masked with the hydrometeor mask developed in this study?

5. P. 3 L. 13-14. It is not clear to what the "*operational* Doppler velocity spectra processing routines" refer to, is this the operational ARSCL processing, the algorithm presented in this study, or perhaps something else?

6. The introduction does not provide enough detail on how insects and hydrometeors have been identified in previous studies to allow the reader to understand the difference in this manuscript.

7. P. 3 L. 9-14. In my opinion, the description of the algorithm here is more detailed than is necessary for an introduction. Instead, the authors could make clear the novelty of their approach compared to previous work.

8. P. 3 L. 17. I suggest to also mention the (quantitative) comparison with the ceilometer measurements here, as they are more substantial for evaluating the performance of the algorithm.

9. P. 4 L 14. There appears to be a problem with the formulation as it seems to me there should not be a Doppler velocity power spectra for each velocity bin. Perhaps the authors mean "The received signals are processed to yield co-polarized [...] and cross-polarized [...] power at each velocity bin $v_i$ and range gate $h_j$", or alternatively "The received signals are processed to yield co-polarized [...] and cross-polarized [...] Doppler velocity power spectra at each range gate $h_j$".

10. P. 4. L. 24-25. Unfortunately I don't understand what is meant by "retrievals for the GE mode available in the ARM data archive". Is the algorithm presented in the paper also applied for the GE mode and made available in the ARM data archive?

11. P. 4 L. 22-35. The citations provided seem all to refer to datasets. The authors should also provide references for technical information on the instrumentation as well as data processing and retrievals applied, where available.

12. P. 4 L. 31. Technical specifications for Doppler lidar are missing: at least the model and wavelength should be mentioned.

13. P. 6 L. 4-5. It is not clear what is suspicious about the lack of LDR above the cloud base.

14. P. 6 L. 24. I did not find the mentioned sequential spectra profiles in the supplemental material.

15. P. 10 L. 2. "By eye" is rather ambiguous, and requires a trained eye from the reader to follow the authors thoughts. The authors could elaborate in text or add labels in the figure to make it clear what they wish to communicate.

16. P. 10 L. 4-8. Sect. 3 describes the characteristics of the radar measurements, specifically LDR and CoPol Doppler spectra, typical for different kind of targets (cloud, precipitation, insects) and forms the basis of understanding the authors reasoning in the following section. I found it confusing to discuss the hydrometeor mask here, before the algorithm has been described. I suggest simply stating that the mask will be discussed later and moving all of the content regarding the mask to a more appropriate section later. Although Fig. 4d could also be moved to a later section, I agree with the authors choice to include the mask in the same figure as the other variables to allow easy comparison.

17. P. 10 L. 6. It is not obvious why it is surprising that the hydrometeor mask is affirmative (after all, the reader does not know how the mask is defined yet), more specifically it is not clear where in the time-height domain the presence of hydrometeors is surprising. I would argue that from about minutes 20 to 40 below 1.5 km it seems rather obvious from Fig. 4a and b that there is nearly continuous precipitation. Since spectra in Sect. 4 is mentioned (P. 10 L. 7-8), I gather that the authors intend to point to the layer below 1.5 km from minutes 45 to 55 (?) as particularly challenging to discriminate between insects and hydrometeors. Clarification needed.

18. P. 10 L. 12. This claim is not supported by what has been shown so far, specifically it has not been shown that examining the texture of the Doppler spectra helps with discriminating between insects and precipitation when precipitation is occurring. I suggest the authors show a spectograph and/or individual spectra for this time (similarly as in Fig. 3 in Sect 3.1.) in Sect. 3.2 and describe which features in the CoPol and LDR spectra suggest the presence of hydrometeors and insects in specific regions of the velocity-range domain. Given the central challenge of separating insects and precipitation when they occur in the same pixel, and the benefit of the evaluation in Doppler velocity domain provides in this regard, I think extending the discussion in Sect. 3.2 would be beneficial.

19. P. 10 L. 19. I believe here should probably be LDR spectra, not XPol?

20. P. 10 L.31-32. "it is assumed that the power in any location ($v_i$, $h_i$) is due to either insect or hydrometeor scattering, and not both." Could the authors comment on the validity of these assumptions, namely that all signal is related to insects or hydrometeors but nothing else, and that both cannot occur at same height and Doppler velocity?

21. P. 10 L.31. Should $h_i$ be $h_j$?

22. P. 12 L. 5-6. "signal power is expressed in decibel units to remove signal magnitude dependencies that occur between cloud droplet (order of 10 dB) and raindrop (order of 50 dB) observations." I cannot follow the authors' logic here, it seems to me that although using decibel units there is a difference in magnitude, 10 vs. 50 dB as stated by the authors.

23. P. 12 L. 19. "texture shown in Fig 6a" - probably a typo, texture is shown in Fig. 6**b**.

24. P. 12 L. 25 and Fig. 7. Why did the authors choose to exclude spectral regions with LDR here? I understand the purpose of the CoPol texture algorithm is to determine whether insects are present when no LDR is available, but I do not see the reason to exclude these data from Fig. 7.

25. Figs. 6d,. 7a,b, 9d, and associated text. I don't see why the authors are showing and discussing STD(texture). From Fig. 7b it is seen, and on P. 13 L. 15 it is stated that STD(texture) and Max(texture) closely correlate, so that no additional information is gained from STD(texture) and it is also not used for the insect-hydrometeor-detection algorithm. I suggest omitting STD(texture) because it plays no role in the described algorithm (if the authors wish they could include a sentence stating that STD(texture) was investigated), or moving the figures and discussion to an appendix.

26. CoPol texture algorithm (Sect. 4.1). Fig. 7b and c do not show clearly separated two populations but considerable overlap in the Max(texture) distributions. Also Luke et al. (2008) mention that differences in the radar Doppler spectrum texture between insects and hydrometeors is sometimes not obvious. The authors should discuss this challenge and its implications for their classification.

27. Sects. 4.1 and 4.2 give the impression that the Max(texture) and Mean(LDR) thresholds for the algorithm were chosen based on one hour of measurements each. Have the authors checked that the chosen thresholds are appropriate beyond the one hour examples?

28. P. 15 L. 1-3 and 12. Should it be *mean[S($v_{i\pm2}$, $h_{j\pm1}$)]* and *STD[S($v_{i\pm2}$, $h_{j\pm1}$)]*?

29. The method to estimate noise the XPol or LDR spectra is not mentioned although it is described for CoPol spectra.

30. Figs. 6h, 9h, 10a,b, and associated text. Similarly as for the texture algorithm (see comment 26), I do not see the need to show and discuss STD(LDR) as it is not utilized in the insect-hydrometeor-detection algorithm.

31. P. 15 L. 2. The authors mention that they only include profiles below 1.5 km in Fig. 10 to avoid ice particle scattering. Is the LDR algorithm only applied for signal below 1.5 km or at temperatures where ice is not expected? If yes, this should be mentioned. If no, why are the authors excluding values above 1.5 km from Fig. 10 if the classification is still performed for regions where ice particle scattering is expected?

32. P. 19 L. 6-7. The authors should give an explanation for the "3-member temporal continuity filter".

33. P. 19 L. 9. I could not find QC2 in the Supplemental Material.

34. P. 20 L. 21-27 and Fig. 13. This comparison between the hydrometeor mask and the ceilometer cloud base height nicely demonstrates the performance of the described algorithm to detect hydrometeors from insects for non-precipitating clouds. Have the authors considered any methods to validate the hydrometeor mask when precipitation occurs, for example by utilizing surface level precipitation measurements?

35. P. 21 L. 6-17 and Fig. 14. Could be omitted. The lidar is similar to ceilometer and does not really provide anything additional to the evaluation of the performance of the algorithm. I also fail to see the value of the total sky imager (TSI) photo for evaluating the algorithm. If the authors wish to show the TSI picture (Fig. 14b) it could be moved to the beginning of the manuscript when the case is introduced.

36. The authors mention that the CoPol spectra texture algorithm allows for filtering insects in the absence of XPol signal. It could also be mentioned that the texture method is applicable for radars without a XPol receiver.

37. P. 23 L. 36-37. The authors should check that the Supplemental Material provided agrees with what is stated in the manuscript.

38. Fig. 3b, d. Is it necessary to show the X-Pol spectograph (Fig. 3b) and example spectra (Fig. 3d)? Fig 3b and d are not discussed in the text, and Fig 3b is near duplicate with Fig. 6e.

39. Fig. 4d and 12b,c. The colorbars seems to have a mistake, the figures look like red is 1 and 0 is white.

40. Fig. 5. What is $ht$ in box 10?

41. Fig. 5 Box 5. Should probably be $S^{XPol}(v_i, h_j)$?

42. Figs. 7c, 10c and 13c. Colorbars unnecessary.

43. Fig. 8a-c and 11a-c. All colors not red or blue could be removed from the colorbars.

44. Fig. 13c. What is the purpose of this figure? Could be omitted.

45. Throughout the manuscript both CoPol (XPol) and Co-Pol (X-Pol) are used, should be unified.

**References**

Luke, E.P, P. Kollias, K.L. Johnson, and E.E. Clothiaux: A Technique for the Automatic Detection of Insect Clutter in Cloud Radar Returns. *J. Atmos. Ocean. Technol.*, 25, 1498-1513, doi:10.1175/2007JTECHA953.1, 2008.

---

## Author Comment (AC1)

**Reply to Reviewer #1**

The authors would like to thank Reviewer #1 for taking the time to read this manuscript and make helpful comments and suggestions. Almost all suggests were incorporated. Detailed replies are below, with *reviewer comments in blue italic Times New Roman font* and author replies in Arial regular font.

Reference to page and line numbers are for the TrackChange document with filename: Williams_kazr_hydro_mask_TrackChange_2021_0505v5.pdf

*In their manuscript „Identifying Insects, Clouds, and Precipitation using Vertically Pointing Polarimetric Radar Doppler Velocity Spectra" Christopher Williams et al describe a combination of spectral texture and LDR to discriminate insects and hydrometeors in cloud radar Doppler spectra. The introduction of the texture and the combination with the frequently used spectral LDR are novel approaches. The topic fits well into the scope of Atmospheric Measurement Techniques and is recommended for publication after minor revisions. Especially the state-of-art review and theoretical basis (scattering properties of insects) can be expanded to provide a more comprehensive overview.*

*Detailed comments*
*1. P1L28: "All datasets and images are available […]". This statement is ambiguous. Are the quicklooks from the dataset available or the figures used in this manuscript?*
There is a delay in making the images available. The lead author (CRW) knew that the daily netCDF data files, TIF images, and animations would be posted on the DOE ARM Archive, but he did not realize that the AMT supplemental material was limited to 50 MB and could not include animations. Paperwork has been submitted requesting DOE ARM archive data, images, and animations for 14 summer months (April-October 2018 and 2019). The netCDF files require 700 MB, the hourly TIF images require about 6 GB, and the profile-by-profile animations require about 5 GB of space. This manuscript will need to wait to be in final form until after DOE approves hosting the archive. In the interim, the netCDF data files, TIF images, and animations are on a Google Drive Folder: https://drive.google.com/drive/folders/1whhYC6op1nzMRg93FJYohjkFB6v1uLNU?usp=sharing
Text was changed to specify that data and images are available on the DOE ARM repository. See text near P1L29.

*2. P2L16: "Due to their large power fluctuations […]". Are the power fluctuations the reason, why insects are detected in the power spectrum?*
Thank you for asking about the power fluctuations because this sentence is not clear. It is not the power fluctuations in the time-domain that enables the insects to be detected in the power spectrum, it is the large change in power between velocity bins (i.e., delta power [dB] per delta velocity [m/s]) that distinguishes insects from clouds. This sentence was modified to improve clarity. See text near P2L16.

*3. P2L26: "[…] of the operational ARSCL processing is identifying and removing insect clutter" More information on the state-of-art in ARSCL is needed. How is insect filtering currently done? What thresholds are used? How does the approach differ from other well established synergistic*

*retrievals, such as Cloudnet? Where are the gaps, this work tries to address? Please also consider showing the ARSCL insect flag for the case studies (Fig 3 and 4).*

Text has been added briefly describing the insect filtering done in the ARSCL products and to address the gaps this work tries to address relative to prior work. Also, the ARSCL hydrometeor-only product was added to Fig. 4. See text near P2L24. See Fig. 4.

*How does your algorithm differ from the MIRA-35 insect filtering, which also relies on spectral LDR (see description in Görsdorf et al. 2015 JAOT and the therein referenced Bauer-Pfundstein and Görsdorf 2007)?*

Thank you for these references to the work of Görsdorf et al. and the work of Bauer-Pfundstein and Görsdorf. The main difference between the proposed method and these works is that the proposed method estimates the scattering process of each pixel in the profile before estimating the location of any peaks. The methods in Bauer-Pfundstein and Görsdorf (as well as Luke et al.) first identify significant peaks in the spectra and then classify each peak as a result of hydrometeor or insect scattering. Text was added to highlight the difference of these two approaches. See text near P4L2.

*4. P4L24: "Figures and algorithm descriptions use the MD mode with retrievals for the GE mode available in the ARM data archive." Does that mean the algorithms for GE are used on the MD data or is this just an additional information that the other mode would be available as well? Please clarify. Does the presented algorithm perform equally well at the GE data?*

When the manuscript was submitted, the plan was to run the same algorithms on the GE mode data and make the GE processed masks available on the ARM data archive. Due to other commitments since submitting this manuscript, the GE mode data has not been processed. Thus, the text was modified to state that the GE mode will be processed in the future and will eventually be available on the ARM archive. See text near P5L5.

*5. P4 Eq. 1b and 2: Is the noise level independently estimated for SXPOL? Please provide the depolarization decoupling of the used system (integrated cross-polarization ratio, Myagkov et al. 2015 JAOT).*

Yes, the noise level of every spectrum is estimated independently for both the CoPol and XPol channels. The KAZR handbook states that the cross polarization isolation is -27 dB. This information and reference to the KAZR handbook were added to the manuscript. See text near P5L8.

*6. P4L26: "The KAZR 0.2° antenna beamwidth [...]". Please give the diameter of the antenna. Is it covered by a radome?*

More information on the KAZR antenna was added to the text. See text near P5L8.

*7. P4L35: Please provide more technical information on the lidars. What are the wavelengths, what algorithms are used for cloud base detection?*

More information on the Vaisala ceilometer was added to the text along with the ARM ceilometer handbook as a reference. See text P5L21.

*8. P5 Table 1: What encoding sequence is used in the MD mode?*
More information about the pulse duration, modulation, range resolution, and spacing between samples has been added to the main text and to Table 1. See text near P5L11. See table near P6L1.

*9. Section 3 (P5L26): reads very phenomenological without addressing the underlying physics of the insect return. Is the observed texture characteristics caused by the point target scattering at individual insects? Such intermittent during the integration time of the Doppler spectrum and easily be identified by a time-frequency analysis of the IQ signal. If you have access to such low-level date it might be worth a look.*
Yes, the high texture, or dP/dv across the power spectrum is caused by individual point targets having a fixed speed during the observation time of about 1.8 seconds. This is in contrast to the distribution of cloud particles moving at different velocities producing a smoother power spectrum. The introductory text to Section 3 was improved to highlight these two scattering processes at the beginning of the section (instead of having this key information hidden deep in the sub-sections.). See text near P6L28.

The authors have requested time-series data from the ARM archive to explore the time-evolution of the I & Q data. After looking at many spectra profiles, the insects tend to occur in two different scenarios: either isolated targets that appear in only one profile, or many targets with different velocities. It will be interesting to look at the data from an entomologist's prospective as the insects move through the radar resolution volume.

*Are the sizes of insects expected at the SGP are similar to the wavelength or much smaller? How does the LDR signature of point targets depend on the pulse shape in the MD mode?*
I (CRW) do not know if LDR changes with pulse shape. I would image that it not because the pulse is transmitted and the received on both CoPol and XPol channels which are decoded with the same compression filter. Changes in the pulse amplitude would affect both channels equally. Maybe this can be explored by comparing the LDR from the MD and GE modes because the GE mode is not using any frequency modulation. Regarding insect size, an insect with diameter smaller than 2.8 mm would still be within the Rayleigh scattering regime.

*10. P6L5: "The abrupt omission of LDR observations above the ceilometer cloud base height appears suspicious as it produces a nearly horizontal feature in Fig. 2d." Can you exclude artefacts of the Doppler spectrum preprocessing, especially noise level estimation in SXPOL and omission of empty spectra in the data?*
The CoPol and XPol spectra are processed with the same code. The code to omit spectra to save disk space is the same for both channels. The code estimating the noise level is the same, also. I (CRW) cannot see how processing could be generating artefacts that suppress the detection of warm clouds. The missing LDR observations fits the logic of spherical warm water cloud particle scattering not producing a cross-polarized return signal. To avoid confusion to the reader, this sentence was deleted. See text near P7L8.

*11. P6L24: I was not able to find the mentioned spectra profiles in the supplement.*
Yes, that was a problem with the lead author did not realizing that there was a size limit (50 MB) and spectra plots were in mp4 animations that are 25 MB per hour file. In order to show the profile-to-profile spectral variability, a random selection of approximately 169 spectra profiles for six different events are now included in the supplement. Also the images and netCDF files being uploaded to the ARM archive are temporarily available on a Google Drive Folder:
https://drive.google.com/drive/folders/1whhYC6op1nzMRg93FJYohjkFB6v1uLNU?usp=sharing.

*12. P6L25: "This indicates that individual insects appear in the […]" What was the horizontal wind speed in that height?*
Good question about horizontal wind. Text was added to the manuscript stating that the surface wind is about 3 m/s and is fast enough to advect passive insects through the radar resolution volume in less than 4 seconds. More analysis is needed to determine whether the insects are actively flying or passively moving with the horizontal wind. See text near P7L29.

*13. P7 Fig 2: Please mark the time of the case study (Fig 3 and 6) in that figure.*
Yes, a vertical dashed line has been added to Fig. 2 and the caption updated. See Fig. 2.

*14. P10L3: "The bottom panel (Fig 3d) […]" Are you referring to Fig 4d?*
Yes, correction made. See text near P11L7.

*15. P10L16: "from insects (including "atmospheric plankton")" Atmospheric plankton other than insects and its signature in cloud radar observations is not mentioned in Section 3.*
Good catch. Atmospheric plankton was added as a possible scattering process in Section 3. See text near P6L26, P6L28, and P6L35.

*16. P11L1: "KAZR XPol channel is not sensitive enough to detect non-precipitating liquid cloud droplets" Liquid non-precipitating clouds should not show any LDR, regardless how sensitive the XPol channel is (at least for a single scattering process).*
Thanks for pointing this out. The text has been improved with your suggestion. See text near P12L3.

*17. P11L5: "[…] that clouds are persistent over 10's of seconds and 10's of meters" But the air velocity, which determines the spectral bin of a signal, might change a smaller timescales. Does this filtering step need to be adjusted to the turbulence conditions?*
Yes, you are correct, the air velocity does have a shorter time scale and shifts the cloud into different velocity bins. But the time-height filtering is being done after the spectral processing steps and is using the binary hydrometeor masks. The text was modified to clarify this filtering step. See text near P12L6.

*18. P11 Fig 5: Please consider adding color coding to clarify the two branches. Also specify 'small regions' more exactly.*

Thank you. Colors and labels were added to Fig. 5 to clarify the two branches. The 'small regions' were explicitly defined. See updated Fig. 5.

*19. P11L9: The heading should read "CoPol Texture Algorithm Branch" to be consistent with 4.2.*

Thank you. Corrected. See text near P12L12.

*20. P11 Eq 5: Is the term in the brackets a matrix or just the maximum out of two alternatives?*

It is the maximum out of two alternatives. Equation 5 has been updated and text was added to clarify the calculation. See text near P13L6 and Eg. 5.

*21. P12L8: "[...] that for radars with broader beamwidths, the insect peak would broaden [...]" Are the broadening processes the same as for distributed targets (e.g. Shupe 2008 JAOTech)?*

Yes, the broadening processes should be the same. Because of the narrow velocity range of the point targets, the radar processing also contributes to the spectrum broadening. The text in this sentence was updated. See text near P13L13.

*22. P12L20 "Interestingly, enhancements in both max texture and STD texture are visible near 1.8 and 2 km indicating that insect scattering is occurring within cloud scattering regions." Judging from Fig 2, this could be a gap in the cloud with low reflectivities and positive vertical velocity, whereas the higher reflectivity around is associated with negative velocities. Likely, 'close proximity' would be a better description.*

Very good observation. Correction was made. See text near P14L2.

*23. P15L13: "A threshold of ... LDR threshold -15 dB clearly separates the two distributions and is indicated with a dashed line in Fig. 10b." This threshold also fits to the findings of Matrosov 1991 (JAS) and Reinking (1997 JAMC).*

Thank you for these references. Text was updated to highlight the work from these two references. See text near P17L10.

*24. P18 Section 4.4: Having a spectrally resolved mask, have you considered calculating the moments of the Doppler spectrum for each population? This could provide further insight into the co-occurrence of insects and clouds in the same volume.*

Yes, we have calculated Doppler velocity spectrum moments for the different masks. We are still evaluating their quality and expect to release the spectral masks and the moments at a later date.

*25. P18L19: The abbreviation QC needs to be defined.*

Corrected. Thank you. See text near P20L19.

*26. P19 Fig 12: What does v06 stand for? If it's just a version control flag, please consider omitting it.*

Yes, it is a version control number. It was removed from the figure. See updated Fig. 12.

*27. P21L14 and P22 Fig 14: What is the additional benefit of the Doppler Lidar observations, that could not be derived from the Ceilometer? From the current content, this part of the comparison could be omitted without loosing information.*

Okay. Figure 14 was deleted and text describing the Doppler Lidar and TSI in section 2 was deleted. See text near P5L18.

*28. P22L13: "[…] cautionary note for future studies is that XPol spectra observations observe fewer insects than CoPol observations". This depends strongly on the signal amplification in the XPol channel and the polarization decoupling of the system.*

Well, that is not what these observations suggest (nor the observations of Matrosov 1991 and Rienking et al 1997). The received XPol signal power will always be less than or equal to the received CoPol signal power (other than random noise fluctuations). This is why LDR has negative values (Fig. 10 shows precipitation with LDR less than -20 dB). The cautionary note is to make the reader aware that less power is returned in the XPol channel than the CoPol channel and that if the reader only uses XPol methods to identify insects in the CoPol channel, then the algorithm will miss insects in the CoPol channel. Text was strengthened to clarify this important finding. See text near P24L12.

*29. P23L4: "[…] many velocity bins and several range gates". Depends on the technical configuration of the radar, please consider providing physical units (m s-1 and m respectively).*

Good point. Typical values are provided to show the stark contrast between point scattering and distributed targets. See text near P25L5.

*30. P23L13: "There appears to be relationships between the insect activity index, radar reflectivity and cloud formation." This is a very vague statement. Please be more concrete or or consider omitting it.*

Omitted. See text near P25L17.

*31. P23L32: Seeing the importance of the issue you are addressing and assuming a broad interest in the community, please consider providing the source code in an open repository similarly to your 2018 paper.*

I think we will make the code available in the near future, after it gets converted into Python. Right now, the code is not well documented and written in MATLAB. The Python version will probably be posted so that users do not need a MATLAB license. There is a comment that the code is available from the lead author upon request.

---

## Author Comment (AC2)

**Reply to Reviewer #2**

The authors would like to thank Reviewer #2 for taking the time to read this manuscript and make helpful comments and suggestions. Most suggests were incorporated. Detailed replies are below, with *reviewer comments in blue italic Times New Roman font* and author replies in Arial regular font.

Reference to page and line numbers are for the TrackChange document with filename: Williams_kazr_hydro_mask_TrackChange_2021_0426v1.pdf (contains changes recommended by reviewer #1 and #2).

*The manuscript titled "Identifying Insects, Clouds, and Precipitation using Vertically Pointing Polarimetric Radar Doppler Velocity Spectra" by Williams et al. describes a two-part algorithm to distinguish radar signal related to hydrometeors or insects in vertically pointing polarimetric Doppler radar measurements. Two techniques, one relying on the morphology of the co-linear (CoPol) Doppler spectrum and the other on the linear depolarization ratio (LDR) specrum, are independently applied and the resulting hydrometeor/insect masks are then combined. The performance of the method is shown by a good agreement between the hydrometeor mask and cloud base height retrieved from ceilometer, which provides an independent observation.*

*The authors make a good effort to illustrate the problem they are addressing (Figs. 1 and 2) as well as the reasoning behind the chosen approaches (Section 3, Figs. 3, 4, 6 and 9). Variables are clearly defined, and except a few small exceptions (see specific comments below) the description of the algorithm is detailed enough to easily follow what has been done. I also appreciate that the authors provide so many cases with the resulting hydrometeor mask in the supplemental material. The authors are addressing a common issue with cloud radar observations and although the concepts behind the described algorithm are not novel, the manuscript provides the community with a practically applicable method. My main concern with the algorithm is related to the choice of thresholds: the authors demonstrate their choice of thresholds (Figs. 7 and 10) using a sub-set of one hour data for two example cases, one case for each algorithm branch, without explaining why the specific sub-set was taken or if the thresholds obtained for the one hour example cases are generalizable to other cases. Furthermore, I find the presentation of the work lacking and needs to be improved before the manuscript can be published.*

*Minor comments*
*1. Page 1 Lines 15-17 (abstract) "The insect-hydrometeor discrimination method uses CoPol and XPol spectral information in two separate algorithms..."*
*This sentence might be misleading because the algorithm does not utilize the XPol spectra as such, but the LDR spectra. Although strictly speaking the LDR is based on "XPol spectral information", I'd suggest to reformulate for clarity.*
As suggested, a sentence was added to the abstract stating that the CoPol and XPol spectra were measured and used to calculate the LDR spectra. See text near P1 L15.

*2. P. 1 L. 25-26. Gives the impression that the hydrometeor mask bottoms are always within +/- 100 m from ceilometer cloud base height when Fig. 13a indicates that this is the case for 71% profiles. The authors should acknowledge here that although most, not all, hydrometeor mask*

*bottoms are within the +/- 100 m from the ceilometer base to accurately reflect the content of the paper in the abstract.*
Good catch. The sentence was modified. See text near P1 L24 and L29.

*3. P. 2 L. 26-27 and Fig. 1. Information on where observations in Fig. 1 are obtained is missing.*
Location of observations added to manuscript body and fig. 1 caption. See text near P2 L32 and P3 L11.

*4. P. 2 L. 27-34 and Fig. 1. The example nicely illustrates the motivation for the study. Have the authors considered also showing in Fig. 1 the reflectivity masked with the hydrometeor mask developed in this study?*
Good idea. The hydrometeor mask developed in this study is now shown in the bottom panel of Fig. 1. See Fig. 1.

*5. P. 3 L. 13-14. It is not clear to what the "operational Doppler velocity spectra processing routines" refer to, is this the operational ARSCL processing, the algorithm presented in this study, or perhaps something else?*
Good suggestion. The text was changed to focus on the algorithm and now indicates that the binary mask is set to true if three or more consecutive velocity bins have hydrometeors. See text near P4, L11.

*6. The introduction does not provide enough detail on how insects and hydrometeors have been identified in previous studies to allow the reader to understand the difference in this manuscript.*
More details about insect and hydrometeor detection in polarimetric and Doppler velocity spectrum methods was added to the introductions. See text near P2 L15, and near P4 L9.

*7. P. 3 L. 9-14. In my opinion, the description of the algorithm here is more detailed than is necessary for an introduction. Instead, the authors could make clear the novelty of their approach compared to previous work.*
The extra detail describing the method was removed from the introduction. Also, a clear distinction from prior work is provided before the example shown in Figure 1. See text near P2 L15, and near P4 L9.

8. P. 3 L. 17. I suggest to also mention the (quantitative) comparison with the ceilometer measurements here, as they are more substantial for evaluating the performance of the algorithm.
Good suggestion. Text was modified. See text near P5 L6.

*9. P. 4 L 14. There appears to be a problem with the formulation as it seems to me there should not be a Doppler velocity power spectra for each velocity bin. Perhaps the authors mean "The received signals are processed to yield co-polarized [...] and cross-polarized [...] power at each velocity bin $v_i$ and range gate $h_j$", or alternatively "The received signals are processed to yield co-polarized [...] and cross-polarized [...] Doppler velocity power spectra at each range gate $h_j$".*
Good catch. I (CRW) was so focused on defining $v_i$ and $h_j$ that I missed the grammar/logic error. Text modified to first suggestion. See text near P5 L25.

10. P. 4. L. 24-25. Unfortunately I don't understand what is meant by "retrievals for the GE mode available in the ARM data archive". Is the algorithm presented in the paper also applied for the GE mode and made available in the ARM data archive?

Yes, it was confusing to mention the GE mode retrievals. The text was modified to focus on the method applied to the MD mode. Also, the GE dataset DOI was removed. See text near P6 L1.

11. P. 4 L. 22-35. The citations provided seem all to refer to datasets. The authors should also provide references for technical information on the instrumentation as well as data processing and retrievals applied, where available.

Good suggestion. The text and references include both the dataset DOI and the technical manual so that readers can find the datasets used in this study. See text near P5 L34 and P6 L17.

12. P. 4 L. 31. Technical specifications for Doppler lidar are missing: at least the model and wavelength should be mentioned.

Per another reviewer's suggestion, the Doppler lidar comparison was removed from this manuscript. But, the model number and wavelength for the ceilometer were added. See text near P6 L17.

13. P. 6 L. 4-5. It is not clear what is suspicious about the lack of LDR above the cloud base.

Per another reviewer's suggestion, this text about suspicious behavior was removed. See text near P8 L8.

14. P. 6 L. 24. I did not find the mentioned sequential spectra profiles in the supplemental material.

Yes, the lead author (CRW) did not realize that there was a size limit (50 MB) in the supplemental material. The spectra plots were in mp4 animations that are 25 MB per hour file. In order to show the profile-to-profile spectral variability, a random selection of approximately 169 spectra profiles for six different events are now included in the supplement. Also the images and netCDF files being uploaded to the ARM archive are temporarily available on a Google Drive Folder: https://drive.google.com/drive/folders/1whhYC6op1nzMRg93FJYohjkFB6v1uLNU?usp=sharing.

15. P. 10 L. 2. "By eye" is rather ambiguous, and requires a trained eye from the reader to follow the authors thoughts. The authors could elaborate in text or add labels in the figure to make it clear what they wish to communicate.

This sentence does not add any value to the manuscript and was removed. Detailed descriptions of reflectivity and LDR calculations are presented later in the manuscript. See text near P12 L 2.

*16. P. 10 L. 4-8. Sect. 3 describes the characteristics of the radar measurements, specifically LDR and CoPol Doppler spectra, typical for different kind of targets (cloud, precipitation, insects) and forms the basis of understanding the authors reasoning in the following section. I found it confusing to discuss the hydrometeor mask here, before the algorithm has been described. I suggest simply stating that the mask will be discussed later and moving all of the content regarding the mask to a more appropriate section later. Although Fig. 4d could also be moved to a later section, I agree with the authors choice to include the mask in the same figure as the other variables to allow easy comparison.*

Good suggestion. The text in this section was moved to section 4. See text near P12 L5 and P23 L5.

*17. P. 10 L. 6. It is not obvious why it is surprising that the hydrometeor mask is affirmative (after all, the reader does not know how the mask is defined yet), more specifically it is not clear where in the time-height domain the presence of hydrometeors is surprising. I would argue that from about minutes 20 to 40 below 1.5 km it seems rather obvious from Fig. 4a and b that there is nearly continuous precipitation. Since spectra in Sect. 4 is mentioned (P. 10 L. 7-8), I gather that the authors intend to point to the layer below 1.5 km from minutes 45 to 55 (?) as particularly challenging to discriminate between insects and hydrometeors. Clarification needed.*

Thank you. The text was modified as suggested. See text near P23 L7.

*18. P. 10 L. 12. This claim is not supported by what has been shown so far, specifically it has not been shown that examining the texture of the Doppler spectra helps with discriminating between insects and precipitation when precipitation is occurring. I suggest the authors show a spectograph and/or individual spectra for this time (similarly as in Fig. 3 in Sect 3.1.) in Sect. 3.2 and describe which features in the CoPol and LDR spectra suggest the presence of hydrometeors and insects in specific regions of the velocity-range domain. Given the central challenge of separating insects and precipitation when they occur in the same pixel, and the benefit of the evaluation in Doppler velocity domain provides in this regard, I think extending the discussion in Sect. 3.2 would be beneficial.*

Good point, the Doppler velocity texture during rain is not shown until Figure 9. The text in this section cannot make conclusions about precipitation classification based on texture until the analysis has been presented. This text was cleaned up in this section and then repeated in the conclusions (Section 6). See text near P12 L11.

*19. P. 10 L. 19. I believe here should probably be LDR spectra, not XPol?*
Yes, XPol changed to LDR. See text near P12 L23.

*20. P. 10 L.31-32. "it is assumed that the power in any location (vi, hi) is due to either insect or hydrometeor scattering, and not both." Could the authors comment on the validity of these assumptions, namely that all signal is related to insects or hydrometeors but nothing else, and that both cannot occur at same height and Doppler velocity?*
Good comment. The text in this section was modified and points the reader to Figs. 6, 8,9, and 11 which show that insects and hydrometeors do occur in the same range gate and at the same velocities. The text was modified to clarified that the CoPol and LDR

algorithms are binary classifiers and overlapping classifications are mitigated by the temporal QC filtering. See text near P12 L 34.

*21. P. 10 L.31. Should hi be hj?*
Yes. Corrected. See text near P12 L35.

*22. P. 12 L. 5-6. "signal power is expressed in decibel units to remove signal magnitude dependencies that occur between cloud droplet (order of 10 dB) and raindrop (order of 50 dB) observations." I cannot follow the authors' logic here, it seems to me that although using decibel units there is a difference in magnitude, 10 vs. 50 dB as stated by the authors.*
Yes, this sentence is confusing…It has been clarified. See text near P14 L 16.

*23. P. 12 L. 19. "texture shown in Fig 6a" - probably a typo, texture is shown in Fig. 6b.*
Yes, that is a typo. Corrected. See text near P15 L10.

*24. P. 12 L. 25 and Fig. 7. Why did the authors choose to exclude spectral regions with LDR here? I understand the purpose of the CoPol texture algorithm is to determine whether insects are present when no LDR is available, but I do not see the reason to exclude these data from Fig. 7.*
Good suggestion. The LDR observations were added to Fig. 7. The Gaussian functional fits to the insect scattering distributions were very similar for distributions with and without LDR signatures. The distributions with LDR show the distributions extending toward the origin (0,0) causing an overlap with the hydrometeor and insect populations. This new figure will be used to address reviewer comment #26 concerning overlap between cloud and insect populations. Figure 7 has been updated. See text near P15 L16.

*25. Figs. 6d,. 7a,b, 9d, and associated text. I don't see why the authors are showing and discussing STD(texture). From Fig. 7b it is seen, and on P. 13 L. 15 it is stated that STD(texture) and Max(texture) closely correlate, so that no additional information is gained from STD(texture) and it is also not used for the insect-hydrometeor-detection algorithm. I suggest omitting STD(texture) because it plays no role in the described algorithm (if the authors wish they could include a sentence stating that STD(texture) was investigated), or moving the figures and discussion to an appendix.*
This comment indicates that the text was too vague and could lead the reader to incorrect conclusions. The text was modified. In short, a correlation between STD(texture) and Max(Texture) does not mean that the STD(texture) does not contain additional information. The high correlation is from the Gaussian function fitting and indicates that the slope of the 2D fitted parameters is close to a 1-to-1 line. (In Fig. 7, note that the contours are slanted at a 45 degree angle.) The magnitude of STD(texture) is different for hydrometeor scattering and insect scattering. The text was modified to use the correlation to develop a texture threshold that is orthogonal to the texture 2D distributions. See text near P16 L8 and text in the new Appendix A.

Given this comment and comment #27, additional analysis was performed and is presented in a new appendix several cloud events and insect events from both years

were used to produce texture and LDR distributions from known cloud and insect observations. There were over 23 million cloud and 24 million insect regions used in the analysis. An appendix was added to show these distributions. Also, the texture threshold is modified to be orthogonal to a line bisecting the cloud and insect distribution centroids exploiting the near 1-to-1 slope of the observations. A receiver operating characteristic (ROC) curve was produced and used to determine the optimal threshold. More details are provided in the text and appendix. See text near P16 L8.

*26. CoPol texture algorithm (Sect. 4.1). Fig. 7b and c do not show clearly separated two populations but considerable overlap in the Max(texture) distributions. Also Luke et al. (2008) mention that differences in the radar Doppler spectrum texture between insects and hydrometeors is sometimes not obvious. The authors should discuss this challenge and its implications for their classification.*

Good suggestion. Figure 7 was modified and Appendix A was added and they show the overlap between hydrometeor and insect scattering distributions. Text was added to describe the overlap and how the time-domain QC filtering discussed in Section 4.4 is one way to remove misclassified observations due to this overlap. See text near P16 L8.

*27. Sects. 4.1 and 4.2 give the impression that the Max(texture) and Mean(LDR) thresholds for the algorithm were chosen based on one hour of measurements each. Have the authors checked that the chosen thresholds are appropriate beyond the one hour examples?*

Appendix A was added showing the representativeness of the distributions using over 75 hours of KAZR observations from 2018 and 2019. The events were hand selected to contain either hydrometeors or insect scattering observations. These 'truth' datasets were used to develop an orthogonal threshold that had a true positive rate of over 0.9. See text near P16 and L8. See Appendix A.

*28. P. 15 L. 1-3 and 12. Should it be mean[S(vi±2, hj±1)] and STD[S(vi±2, hj±1)]?*
Corrected. See text near P17 L14.

*29. The method to estimate noise the XPol or LDR spectra is not mentioned although it is described for CoPol spectra.*
Text was added describing the XPol spectra noise estimate method. See text near P17 L6.

*30. Figs. 6h, 9h, 10a, b, and associated text. Similarly as for the texture algorithm (see comment 26) I do not see the need to show and discuss STD(LDR) as it is not utilized in the insect-hydrometeor detection algorithm.*
Yes, it is more work to include the STD(LDR), but the STD(LDR) is needed to understand how hydrometeor and insect scattering separate into two different distributions in the LDR domain.

*31. P. 15 L. 2. The authors mention that they only include profiles below 1.5 km in Fig. 10 to avoid ice particle scattering. Is the LDR algorithm only applied for signal below 1.5 km or at temperatures where ice is not expected? If yes, this should be mentioned. If no, why are the*

authors excluding values above 1.5 km from Fig. 10 if the classification is still performed for regions where ice particle scattering is expected?

The choice of limiting profiles to below 1.5 km was to limit the number of hydrometeor observations so that the insect scattering observations would appear in the plot. The text was modified to clarify this graphical limitation (it was not an ice scattering issue). See text near P18 L9.

*32. P. 19 L. 6-7. The authors should give an explanation for the "3-member temporal continuity filter".*

Explained. See text near P22 L7.

*33. P. 19 L. 9. I could not find QC2 in the Supplemental Material.*

The QC2 images will be available on the ARM Archive. The text was modified. See text near P22 L9.

*34. P. 20 L. 21-27 and Fig. 13. This comparison between the hydrometeor mask and the ceilometer cloud base height nicely demonstrates the performance of the described algorithm to detect hydrometeors from insects for non-precipitating clouds. Have the authors considered any methods to validate the hydrometeor mask when precipitation occurs, for example by utilizing surface level precipitation measurements?*

The only validation of this work has been by examining individual spectra. There is not an independent observational data set that can be used to verify this work.

*35. P. 21 L. 6-17 and Fig. 14. Could be omitted. The lidar is similar to ceilometer and does not really provide anything additional to the evaluation of the performance of the algorithm. I also fail to see the value of the total sky imager (TSI) photo for evaluating the algorithm. If the authors wish to show the TSI picture (Fig. 14b) it could be moved to the beginning of the manuscript when the case is introduced.*

Figure 14 has been removed.

*36. The authors mention that the CoPol spectra texture algorithm allows for filtering insects in the absence of XPol signal. It could also be mentioned that the texture method is applicable for radars without a XPol receiver.*

Text was added to the conclusion section. See text near P26 L10.

*37. P. 23 L. 36-37. The authors should check that the Supplemental Material provided agrees with what is stated in the manuscript.*

Text was modified. See text near P26 L30.

*38. Fig. 3b, d. Is it necessary to show the X-Pol spectograph (Fig. 3b) and example spectra (Fig. 3d)? Fig 3b and d are not discussed in the text, and Fig 3b is near duplicate with Fig. 6e.*

Showing simultaneous CoPol and XPol spectra profiles show that the XPol channel does not observe all of the insects. It highlights that XPol will not identify all of the insects observed in the CoPol channel.

*39. Fig. 4d and 12b,c. The colorbars seems to have a mistake, the figures look like red is 1 and 0 is white.*
Colorbars are removed.

*40. Fig. 5. What is ht in box 10?*
Removed.

*41. Fig. 5 Box 5. Should probably be SXPol(vi, hj)?*
Corrected.

*42. Figs. 7c, 10c and 13c. Colorbars unnecessary.*
Corrected.

43. Fig. 8a-c and 11a-c. All colors not red or blue could be removed from the colorbars.
Corrected.

*44. Fig. 13c. What is the purpose of this figure? Could be omitted.*
*Removed.*

*45. Throughout the manuscript both CoPol (XPol) and Co-Pol (X-Pol) are used, should be unified.*
Corrected.

*References*
*Luke, E.P, P. Kollias, K.L. Johnson, and E.E. Clothiaux: A Technique for the Automatic Detection of Insect Clutter in Cloud Radar Returns. J. Atmos. Ocean. Technol., 25, 1498-1513, doi:10.1175/2007JTECHA953.1, 2008.*
Included in body of text.